# Exact Upper and Lower Bounds for the Output Distribution
# of Neural Networks with Random Inputs

**Andrey Kofnov** [1]  **Daniel Kapla** [1]  **Ezio Bartocci** [2]  **Efstathia Bura** [1]

## Abstract

We derive exact upper and lower bounds for the cumulative distribution function (cdf) of the output of a neural network (NN) over its entire support subject to noisy (stochastic) inputs. The upper and lower bounds converge to the true cdf over its domain as the resolution increases. Our method applies to any feedforward NN using continuous monotonic piecewise twice continuously differentiable activation functions (e.g., ReLU, tanh and softmax) and convolutional NNs, which were beyond the scope of competing approaches. The novelty and instrumental tool of our approach is to bound general NNs with ReLU NNs. The ReLU NN-based bounds are then used to derive the upper and lower bounds of the cdf of the NN output. Experiments demonstrate that our method delivers guaranteed bounds of the predictive output distribution over its support, thus providing exact error guarantees, in contrast to competing approaches.

## 1. Introduction

Increased computational power, availability of large datasets, and the rapid development of new NN architectures contribute to the ongoing success of NN based learning in image recognition, natural language processing, speech recognition, robotics, strategic games, etc. A limitation of NN machine learning (ML) approaches is that they lack a built-in mechanism to assess the uncertainty or trustworthiness of their predictions, especially on unseen or out-of-distribution data. A NN is a model of the form:

$$Y = f(\mathbf{X}, \mathbf{\Theta}), \tag{1}$$

where $Y$ is the output and $\mathbf{X}$ the input (typically multivariate), and $f$ is a *known* function modeling the relationship between $\mathbf{X}$ and $Y$ parametrized by $\mathbf{\Theta}$. Model (1) incorporates uncertainty neither in $Y$ nor in $\mathbf{X}$ and NN fitting is a numerical algorithm for minimizing a loss function. Lack of uncertainty quantification, such as assessment mechanisms for prediction accuracy beyond the training data, prevents neural networks, despite their potential, from being deployed in safety-critical applications ranging from medical diagnostic systems (e.g., (Hafiz & Bhat, 2020)) to cyber-physical systems such as autonomous vehicles, robots or drones (e.g., (Yurtsever et al., 2020)). Also, the deterministic nature of NNs renders them highly vulnerable to not only adversarial noise but also to even small perturbations in inputs ((Bibi et al., 2018; Fawzi et al., 2018; Goodfellow et al., 2014; 2015; Hosseini et al., 2017)).

Uncertainty in modeling is typically classified as *epistemic* or *systematic*, which derives from lack of knowledge of the model, and *aleatoric* or *statistical*, which reflects the inherent randomness in the underlying process being modeled (see, e.g., (Hüllermeier & Waegeman, 2021)). The *universal approximation theorem* (UAT) (Cybenko, 1989; Hornik et al., 1989) states that a NN with one hidden layer can approximate any continuous function for inputs within a specific range by increasing the number of neurons. In the context of NNs, epistemic uncertainty is of secondary importance to aleatoric uncertainty. Herein, we focus on studying the effect of random inputs on the output distribution of NNs and derive uniform upper and lower bounds for the cdf of the outputs of a NN subject to noisy (stochastic) input data.

We evaluate our proposed framework on four benchmark datasets (Iris (Fisher, 1936), Wine (Aeberhard & Forina, 1992), Diabetes (Efron et al., 2004), and Banana (Jaichandaran, 2017)), and demonstrate the efficacy of our approach to bound the cdf of the NN output subject to Gaussian and Gaussian mixture inputs. We demonstrate that our bounds cover the true underlying cdf over its entire support. In contrast, the similar but approximate approach of Krapf et al. (2024), as well as high-sample Monte-Carlo simulations, produce estimates outside the bounds over areas of the output range where the bounds are tight.

[1]Faculty of Mathematics and Geoinformation, TU Wien, Vienna, Austria [2]Faculty of Informatics, TU Wien, Vienna, Austria. Correspondence to: Andrey Kofnov <andrey.kofnov@tuwien.ac.at>.

*Proceedings of the $42^{nd}$ International Conference on Machine Learning*, Vancouver, Canada. PMLR 267, 2025. Copyright 2025 by the author(s).

## 2. Statement of the Problem

A NN is a mathematical model that produces outputs from inputs. The input is typically a vector of predictor variables, $\mathbf{X} \in \mathbb{R}^{n_0}$, and the output $Y$, is univariate or multivariate, continuous or categorical.

A *feedforward NN* with $L$ layers from $\mathbb{R}^{n_0} \to \mathbb{R}^{n_L}$ is a composition of $L$ functions,

$$f_L(\mathbf{x} \mid \boldsymbol{\Theta}) = f^{(L)} \circ f^{(L-1)} \circ \ldots \circ f^{(1)}(\mathbf{x}), \qquad (2)$$

where the $l$-th layer is given by

$$f^{(l)}(\mathbf{x} \mid \mathbf{W}^{(l)}, \mathbf{b}^{(l)}) = \sigma^{(l)}(\mathbf{W}^{(l)}\mathbf{x} + \mathbf{b}^{(l)}),$$

with weights $\mathbf{W}^{(l)} \in \mathbb{R}^{n_l \times n_{l-1}}$, bias terms $\mathbf{b}^{(l)} \in \mathbb{R}^{n_l}$, and a non-constant, continuous activation function $\sigma^{(l)} : \mathbb{R} \to \mathbb{R}$ that is applied component-wise. The NN parameters are collected in $\boldsymbol{\Theta} = (\text{vec}(\mathbf{W}_1), \mathbf{b}_1, \ldots, \text{vec}(\mathbf{W}_L), \mathbf{b}_L) \in \mathbb{R}^{\sum_{l=1}^{L}(n_{l-1} \cdot n_l + n_l)}$. [1] The first layer that receives the input $\mathbf{x}$ is called the *input layer*, and the last layer is the *output layer*. All other layers are called *hidden*. For categorical outputs, the class label is assigned by a final application of a decision function, such as $\arg\max$.

Despite not being typically acknowledged, the training data in NNs are drawn from larger populations, and hence they contain only limited information about the corresponding population. We incorporate the uncertainty associated with the observed data assuming that they are random draws from an unknown distribution of bounded support. That is, the data are comprised of $m$ draws from the joint distribution of $(\mathbf{X}, Y)$, and the network is trained on observed $(\mathbf{x}_i, y_i)$, $\mathbf{x}_i = (x_{i1}, x_{i2}, \ldots, x_{in_0})$, and $y_i$, $i = 1, \ldots, m$. [2] A NN with $L$ layers and $n_l$ neurons at each layer, $l = 1, \ldots, L$, is trained on the observed $(\mathbf{x}_i, y_i)$, $i = 1, \ldots, m$, to produce $m$ outputs $\tilde{y}_i$, and the vector of the NN parameters, $\boldsymbol{\Theta} = (\text{vec}(\mathbf{W}_1), \mathbf{b}_1, \ldots, \text{vec}(\mathbf{W}_L), \mathbf{b}_L)$, is obtained. $\boldsymbol{\Theta}$ uniquely identifies the trained NN. Given $\boldsymbol{\Theta}$, we aim to quantify the robustness of the corresponding NN, to perturbations in the input variables. For this, we let

$$\mathbf{X} \sim F_{\mathbf{X}}, \qquad (3)$$

where $\mathbf{X} \in \mathbb{R}^{n_0}$ stands for the randomly perturbed input variables with cdf $F_{\mathbf{X}}$ and probability density function (pdf) $\phi(\mathbf{x})$ that is piecewise continuous and bounded on a compact support. We study the *propagation of uncertainty* (effect of the random perturbation) in the NN by deriving upper and lower bounds of the cdf $F_{\widetilde{\mathbf{Y}}}(y) = \mathbb{P}(\widetilde{\mathbf{Y}} \leq y)$ of the *random output*, $\widetilde{\mathbf{Y}} = f_L(\mathbf{X} \mid \boldsymbol{\Theta})$. [3]

**Our contributions:**

1. We develop a method to compute the exact cdf of the output of ReLU NNs with random input pdf, which is a piecewise polynomial over a compact hyperrectangle. This result, which can be viewed as a stochastic analog to the Stone-Weierstrass theorem, [4] significantly contributes to the characterization of the distribution of the output of NNs with piecewise-linear activation functions under any input continuous pdf.

2. We derive *guaranteed* upper and lower bounds of the NN output distribution resulting from random input perturbations on a fixed support. This provides *exact* upper and lower bounds for the output cdf provided the input values fall within the specified support. No prior knowledge about the true output cdf is required to guarantee the validity of our bounds.

3. We show the convergence of our bounds to the true cdf; that is, our bounds can be refined to arbitrary accuracy.

4. We provide a constructive proof that any feedforward NN with continuous monotonic *piecewise twice continuously differentiable* [5] activation functions can be approximated from above and from below by a fully connected ReLU network, achieving any desired level of accuracy. Moreover, we enable the incorporation of multivariate operations such as $\max$, $\text{product}$ and $\text{softmax}$, as well as some non-monotonic functions such as $|x|$ and $x^n$, $n \in \mathbb{N}$.

5. We prove a new *universal **distribution** approximation theorem* (UDAT), which states that we can estimate the cdf of the output of any continuous function of a random variable (or vector) that has a continuous distribution supported on a compact hyperrectangle, achieving any desired level of accuracy.

## 3. Our Approximation Approach

We aim to estimate the cdf $F_{\widetilde{\mathbf{Y}}}(y)$ of the output $\widetilde{\mathbf{Y}} = f_L(\mathbf{X} \mid \boldsymbol{\Theta})$ of the NN in (2) under (3); i.e., subject to random perturbations of the input $\mathbf{X}$. We do so by computing upper and lower bounds of $F_{\widetilde{\mathbf{Y}}}$; that is, we compute $\overline{F}_{\widetilde{\mathbf{Y}}}, \underline{F}_{\widetilde{\mathbf{Y}}}$ such that

$$\underline{F}_{\widetilde{\mathbf{Y}}}(y) \leq F_{\widetilde{\mathbf{Y}}}(y) \leq \overline{F}_{\widetilde{\mathbf{Y}}}(y), \ \forall y \qquad (4)$$

We refer to the NN in (2) as *prediction NN* when needed for clarity. We estimate the functions $\overline{F}_{\widetilde{\mathbf{Y}}}, \underline{F}_{\widetilde{\mathbf{Y}}}$ on a "superset" of the output domain of the prediction NN (2) via an

---

[1] The operation $\text{vec} : \mathbb{R}^{n_{l-1} \times n_l} \to \mathbb{R}^{n_{l-1} \cdot n_l}$ stacks the columns of a matrix one after another.

[2] We use the convention of denoting random quantities with capital letters and their realizations (observed) by lowercase letters.

[3] The notation $f_L(\mathbf{X} \mid \boldsymbol{\Theta})$ signifies that $\boldsymbol{\Theta}$, equivalently the NN, is fixed and only $\mathbf{X}$ varies.

[4] A significant corollary to the Stone-Weierstrass theorem is that any continuous function defined on a compact set can be uniformly approximated as closely as desired by a polynomial.

[5] A wide class of the most common continuous activation functions, including ReLU, $\tanh$ and logistic function.

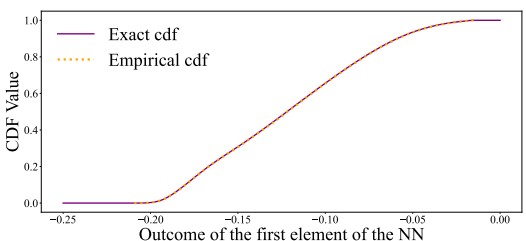

**Figure 1:** Exact cdf of the ReLU NN outcome for class *Setosa* in the Iris data, assuming Beta-distributed inputs.

integration procedure. The cdf of $\widetilde{\mathbf{Y}}$ is given by

$$F_{\widetilde{\mathbf{Y}}}(y) = \mathbb{P}(\widetilde{\mathbf{Y}} \leq y) = \int_{\{\widetilde{\mathbf{Y}} \leq y\}} \phi(\mathbf{x})d\mathbf{x}, \qquad (5)$$

where $\phi(\mathbf{x})$ is the pdf of $\mathbf{X}$. To bound $F_{\widetilde{\mathbf{Y}}}$, we bound $\phi$ by its upper $\overline{\phi}$ and lower $\underline{\phi}$ estimates on the bounded support of $\phi$ as described in Section 3.3. If $\phi$ is a piecewise polynomial, then (5) can be computed exactly for a ReLU prediction network, as we show in Section 3.1. Once $\overline{\phi}$ and $\underline{\phi}$ are estimated, then

$$\underline{F}_{\widetilde{\mathbf{Y}}}(y) = \int_{\{\widetilde{\mathbf{Y}} \leq y\}} \underline{\phi}(\mathbf{x})dx \leq F_{\widetilde{\mathbf{Y}}}(y) \leq \qquad (6)$$

$$\int_{\{\widetilde{\mathbf{Y}} \leq y\}} \overline{\phi}(\mathbf{x})dx = \overline{F}_{\widetilde{\mathbf{Y}}}(y)$$

*Remark* 3.1. $\underline{F}_{\widetilde{\mathbf{Y}}}(y)$ and $\overline{F}_{\widetilde{\mathbf{Y}}}(y)$ are not always true cdfs since we allow the lower estimator not to achieve 1, while the upper bound is allowed to take the smallest value greater than 0.

## 3.1. Exact cdf evaluation for a fully connected NN with ReLU activation function

**Definition 3.2** (Almost disjoint sets). We say that sets $\mathcal{A}$ and $\mathcal{B}$ are almost disjoint with respect to measure $\alpha$, if $\alpha(\mathcal{A} \cap \mathcal{B}) = 0$.

**Definition 3.3** (Closed halfspace). A $n_0$-dimensional closed halfspace is a set $H = \{\mathbf{x} \in \mathbb{R}^{n_0} | \mathbf{v}^T \mathbf{x} \leq c\}$ for $c \in \mathbb{R}$ and some $\mathbf{v} \in \mathbb{R}^{n_0}$ called the normal of the halfspace.

It is known that a convex polytope can be represented as an intersection of halfspaces, called $\mathcal{H}$-representation (Ziegler, 1995).

**Definition 3.4** ($\mathcal{H}$-polytope). A $n_0$-dimensional $\mathcal{H}$-polytope $\mathcal{P} = \bigcap_{j=1}^{h} H_i$ is the intersection of finitely many closed halfspaces.

**Definition 3.5** (Simplex). A $n_0$-dimensional simplex is a $n_0$-dimensional polytope with $n_0 + 1$ vertices.

**Definition 3.6** (Piecewise polynomial). A function $p : K \to \mathbb{R}^{n_L}$ is a *piecewise polynomial*, if there exists a finite set of $n_0$-simplices such that $K = \bigcup_{i=1}^{q} k_i$ and the function $p$ constrained to the interior $k_i^{\circ}$ of $k_i$ is a polynomial; that is, $p|_{k_i^{\circ}} : k_i^{\circ} \to \mathbb{R}^{n_L}$ is a polynomial for all $i = 1, \ldots, q$.

*Remark* 3.7. We do not require piecewise polynomials to be continuous everywhere on the hyperrectangle. Specifically, we allow discontinuities at the borders of simplices. However, the existence of left and right limits of the function at every point on the bounded support is guaranteed by properties of polynomials.

(Raghu et al., 2017) showed that ReLU deep networks divide the input domain into activation patterns (see (Sudjianto et al., 2020)) that are disjoint convex polytopes $\{\mathcal{P}_j\}$ over which the output function is locally represented as the affine transformation $f_L(\mathbf{x}) = NN^j(\mathbf{x}) = \mathbf{c}^j + \mathbf{V}^j \mathbf{x}$ for $\mathbf{x} \in \{\mathcal{P}_j\}$, the number of which grows at the order $\mathcal{O}((\max\{n_l\}_{l=1,\ldots,L})^{n_0 L})$. (Sudjianto et al., 2020) outline an algorithm for extracting the full set of polytopes and determining local affine transformations, including the coefficients $\mathbf{c}^j, \mathbf{V}^j$ for all $\{\mathcal{P}_j\}$, by propagating through the layers. For our computations, we utilize a recent GPU-accelerated algorithm from (Berzins, 2023).

We aim to derive a superset of the range of the network output. For this, we exploit the technique of Interval Bound Propagation (IBP), based on ideas from (Gowal et al., 2019; Wang et al., 2022; Gehr et al., 2018). Propagating the $n_0$-dimensional box through the network leads to the superset of the range of the network output. We compute the cdf of the network's output at each point of a grid of the superset of the output range.

**Theorem 3.8** (Exact cdf of ReLU NN w.r.t. piecewise polynomial pdf). *Let $\widetilde{\mathbf{Y}} : \mathbb{R}^{n_0} \to \mathbb{R}^{n_L}$ be a feedforward ReLU NN, which splits the input space into a set of almost disjoint polytopes $\{\mathcal{P}_j\}_{j=1}^{q_Y}$ with local affine transformations $\widetilde{\mathbf{Y}}(\mathbf{x}) = NN^j(\mathbf{x})$ for $\mathbf{x} \in \mathcal{P}_j$. Let $\phi(\mathbf{x})$ denote the pdf of the random vector $\mathbf{X}$ that is a piecewise polynomial with local polynomials, $\phi(\mathbf{x}) = \phi_i(\mathbf{x})$ for all $\mathbf{x} \in k_i^{\circ}$ over an almost disjoint set of simplices $\{k_i\}_{i=1}^{q_\phi}$, and a compact hyperrectangle support $K \subset \mathbb{R}^{n_0}$. Then, the cdf of $\widetilde{\mathbf{Y}}$ at point $\mathbf{y} \in \mathbb{R}^{n_L}$ is*

$$F_{\widetilde{\mathbf{Y}}}(\mathbf{y}) = \mathbb{P}\left[\widetilde{\mathbf{Y}} \leq \mathbf{y}\right] = \sum_{i=1}^{q_\phi} \sum_{j=1}^{q_Y} \mathcal{I}\left[\phi_i(\mathbf{x}); \mathcal{P}_{j,i}^r\right]$$

$$= \sum_{i=1}^{q_\phi} \sum_{j=1}^{q_Y} \sum_{s=1}^{S_{i,j}} \mathcal{I}\left[\phi_i(\mathbf{x}); \mathcal{T}_{i,j,s}\right],$$

*where $\mathcal{I}\left[\phi_i(\mathbf{x}); \mathcal{T}_{i,j,s}\right]$ is the integral of the polynomial $\phi_i(\mathbf{x})$ over the simplex $\mathcal{T}_{i,j,s}$ such that the reduced poly-*

*tope*

$$\mathcal{P}_{j,i}^r = \mathcal{P}_j \cap k_i \cap \{\mathbf{x} : NN^j(\mathbf{x}) \le \mathbf{y}\} = \bigcup_{s=1}^{S_{i,j}} \mathcal{T}_{i,j,s} \quad (7)$$

*is defined by the intersection of polytopes $\mathcal{P}_j$ and $k_i$, and the intersection of halfspaces*

$$\{\mathbf{x} : NN^j(\mathbf{x}) \le \mathbf{y}\} = \bigcap_{t=1}^{n_L} \left\{\mathbf{x} : NN_t^j(\mathbf{x}) \le y_t\right\}.$$

Theorem 3.8 is shown in Appendix A.1. The proof relies on the algorithm for evaluating the integral of a polynomial over a simplex, as described in (Lasserre, 2021). The right-hand side of (7) results from the Delaunay triangulation (Delaunay, 1934), dividing the reduced polytope $\mathcal{P}_{j,i}^r$ into $S_{i,j}$ almost disjoint simplices $\{\mathcal{T}_{i,j,s}\}$.

*Remark* 3.9. Theorem 3.8 is a tool for approximating the output cdf of any feedforward NN with piecewise linear activation functions on a compact domain, given random inputs of arbitrary continuous distribution at any desired degree of accuracy.[4]

*Example* 3.10. We compute the output cdf of a 3-layer, 12-neuron fully connected ReLU NN with the last (before $\mathrm{softmax}$) linear 3-neuron layer trained on the Iris dataset (Fisher, 1936). The Iris dataset consists of 150 samples of iris flowers from three different species: Setosa, Versicolor, and Virginica. Each sample includes four features: Sepal Length, Sepal Width, Petal length, and Petal width. We focus on two features, *Sepal Length* and *Sepal Width* (scaled to $[0, 1]$), to classify flowers into three classes. Specifically, we recover the distribution of the first component (class *Setosa*) before applying the $\mathrm{softmax}$ function, assuming Beta-distributed inputs with parameters $(2, 2)$ and $(3, 2)$. The exact cdf is plotted in purple in Figure 1, with additional details provided in Appendix D. The agreement with the empirical cdf is almost perfect.

## 3.2. Algorithm for Upper and Lower Approximation of the Neural Network using ReLU activation functions.

We start by showing a general result in Theorem 3.11.

**Theorem 3.11.** *Let $\widetilde{Y}$ be a feedforward NN with $L$ layers of arbitrary width with continuous activation functions. There exist sequences of fully connected ReLU NNs $\{\overline{Y}_n\}$, $\{\underline{Y}_n\}$, which are monotonically decreasing and increasing, respectively, such that for any $\epsilon > 0$ and any compact hyperrectangle $K \subset \mathbb{R}^{n_0}$, there exists $N \in \mathbb{N}$ such that for all $n \ge N$*

$$0 \le \widetilde{Y}(\mathbf{x}) - \underline{Y}_n(\mathbf{x}) < \epsilon, \quad 0 \le \overline{Y}_n(\mathbf{x}) - \widetilde{Y}(\mathbf{x}) < \epsilon$$

*for all $\mathbf{x} \in K$.*

The proof is presented in Appendix A.2.

Theorem 3.11 cannot be directly applied in practice. We develop an approach to approximate the activation functions of a NN provided they are non-decreasing piecewise twice continuously differentiable.

**Definition 3.12.** Let $\Omega = [\underline{a}, \overline{a}] \subset \mathbb{R}$ be a closed interval and $g : \Omega \to \mathbb{R}$ is well-defined, continuous on $\Omega$. We will say that $g$ is **piecewise twice continuously differentiable** on $\Omega$, that is $g \in \mathcal{C}_{p.w.}^2(\Omega)$, if there exists a finite partition of $\Omega$ into closed subintervals $\bigcup_{i=1}^n [a_i, a_{i+1}] = \Omega$ where $\underline{a} = a_1 < a_2 < \ldots < a_{n+1} = \overline{a}$ and $n \in \mathbb{N}$, such that $g\big|_{[a_i, a_{i+1}]}$ is twice continuously differentiable.

Our method provides a **constructive** proof for the restricted case of Theorem 3.11 for non-decreasing activation functions in $\mathcal{C}_{p.w.}^2(\Omega_i)$ at each node $i$ of a NN, where $\Omega_i$ is the input domain of node $i$, and applies to any fully connected or convolutional (CNN) feedforward NN with such activation functions analyzed on a hyperrectangle. The key features of our approach are:

- **Local adaptability:** The algorithm adapts to the curvature of the activation function, providing an adaptive approximation scheme depending on whether the function is locally convex or concave.

- **Streamlining:** By approximating the network with piecewise linear functions, the complexity of analyzing the network output is significantly reduced.

How it works:

**Input/Output range evaluation:** Using IBP (Gowal et al., 2019), we compute supersets of the input and output ranges of the activation function for every neuron and every layer.

**Segment Splitting:** First, input intervals are divided into macro-areas based on inflection points (different curvature areas) and points of discontinuity in the first or second derivative (e.g., 0 for ReLU). Next, these macro-areas are subdivided into intervals based on user-specified points or their predefined number within each range. The algorithm utilizes knowledge about the behavior of the activation function and differentiates between concave and convex regions of the activation function, which impacts how the approximations are constructed and how to choose the points of segment splitting. A user defines the number of splitting segments and the algorithm ensures the resulting disjoint sub-intervals are properly ordered and on each sub-interval the function is either concave or convex. If the function is linear in a given area, it remains unchanged, with the upper and lower approximations equal to the function itself.

**Upper and Lower Approximations:** The method constructs tighter upper and lower bounds through the specific choice of points and subsequent linear interpolation. It calculates new points within each interval (one per interval) and uses them to refine the approximation, ensuring the linear segments closely follow the curvature of the activation function.

Our method differentiates between upper and lower approximations over concave and convex segments. For upper (lower) approximations on convex (concave) segments $[a_k, a_{k+1}]$, we employ local linear interpolation by introducing an intermediate point $a_{k'}$ between the segment endpoints. That is, we let

$$a_{k'} = a_{k'}^{lin\_int}, \ a_{k'} \in [a_k, a_{k+1}]$$
$$\widetilde{f}(x) = \widetilde{f}^{lin\_int}(x) \ \text{for} \ x \in [a_k, a_{k+1}]$$

Conversely, for upper (lower) approximations on concave (convex) segments $[a_k, a_{k+1}]$, we construct a piecewise tangent approximation by inserting a linking point $a_{k'}$ between the function tangent at the segment boundaries. That is,

$$a_{k'} = a_{k'}^{pie\_tan}, \ a_{k'} \in [a_k, a_{k+1}]$$
$$\widetilde{f}(x) = \widetilde{f}^{pie\_tan}(x) \ \text{for} \ x \in [a_k, a_{k+1}]$$

The method guarantees that the piecewise linear approximation of the activation function for each neuron (a) is a non-decreasing function, and (b) the output domain remains the same.

To see this, consider a layer neuron. For upper (lower) approximation on a convex (concave) segment, we choose a midpoint $a_{k'} = (a_k + a_{k+1})/2$ for each subinterval $[a_k, a_{k+1}]$ and compute a linear interpolation, as follows:

$$\kappa_1 = \frac{f(a_{k'}) - f(a_k)}{a_{k'} - a_k}, \quad \kappa_2 = \frac{f(a_{k+1}) - f(a_{k'})}{a_{k+1} - a_{k'}},$$
$$\widetilde{f}(\tau) = f(a_k) + (\tau - a_k)\kappa_1, \quad \tau \in [a_k, a_{k'}]$$
$$\widetilde{f}(\tau) = f(a_{k'}) + (\tau - a_{k'})\kappa_2, \quad \tau \in [a_{k'}, a_{k+1}]$$

For upper (lower) approximation on a concave (convex) segment, we compute derivatives and look for tangent lines at border points of the sub-interval $[a_k, a_{k+1}]$. We choose a point $a_{k'} : a_k \leq a_{k'} \leq a_{k+1}$ to be the intersection of the tangent lines. The original function is approximated by the following two tangent line segments:

$$a_{k'} = \frac{f(a_k) - f(a_{k+1}) - (f'_+(a_k)a_k - f'_-(a_{k+1})a_{k+1})}{f'_-(a_{k+1}) - f'_+(a_k)}$$
$$\widetilde{f}(\tau) = f(a_k) + f'_+(a_k)(\tau - a_k), \quad \tau \in [a_k, a_{k'}]$$
$$\widetilde{f}(\tau) = f(a_{k+1}) + f'_-(a_{k+1})(\tau - a_{k+1}), \ \tau \in [a_{k'}, a_{k+1}],$$

where $f'_-(\cdot), f'_+(\cdot)$ are left and right derivatives, respectively.

For monotonically increasing functions, this procedure guarantees that the constructed approximators are exact upper and lower approximations. Moreover, decreasing the step size (increasing the number of segments) reduces the error at each point, meaning that the sequences of approximators for the activation functions at each node are monotonic: $\overline{f}_{n+1}(x) \leq \overline{f}_n(x), \underline{f}_{n+1}(x) \geq \underline{f}_n(x)$, for all $x$ in the IBP domain. This procedure of piecewise linear approximation of each activation function at each neuron is equivalent to forming a one-layer ReLU approximation network for the given neuron.

By the UAT, for any continuous activation function $\sigma_{l,i}$ at each neuron and any positive $\epsilon_{l,i}$ we can always find a ReLU network-approximator $NN_{l,i}$, such that $|NN_{l,i}(x) - \sigma_{l,i}(x)| < \epsilon_{l,i}$ for all $x$ in the input domain, defined by the IBP. To find such an approximating network we need to choose the corresponding number of splitting segments of the IBP input region. Uniform convergence is preserved by Dini's theorem, which states that a monotonic sequence of continuous functions that converges pointwise on a compact domain to a continuous function also converges uniformly. Moreover, the approximator always stays within the range of the limit function, ensuring that the domain in the next layer remains unchanged and preserves its uniform convergence.

---

**Algorithm 1** Piecewise Linear U/L Approximation

1: **INPUT:** Interval bounds $[\underline{a}_\eta, \overline{a}_\eta]$ and activation function $f$ for neuron $\eta$
2: **OUTPUT:** Sets $\{[a_k, a_{k+1}]\}$, $\{\overline{f}|_k(x)\}$, $\{\underline{f}|_k(x)\}$
3: **PROCEDURE:**
4: Split $[\underline{a}_\eta, \overline{a}_\eta]$ into segments $\{[a_k, a_{k+1}]\}$ such that the $f|_k : [a_k, a_{k+1}] \rightarrow \mathbb{R}$ with $f|_k(x) = f(x)$ is either linear **or** twice continuously differentiable with constant sign of $f|_k''$
5: **for** each segment $[a_k, a_{k+1}]$ **do**
6:     **if** $f|_k$ is linear **then**
7:         $\overline{f}|_k(x) = \underline{f}|_k(x) = f|_k(x)$
8:     **else if** $f|_k'' \geq 0$ **then**
9:         $\overline{f}|_k(x) = \widetilde{f}^{lin\_int}|_k(x)$
10:         $\underline{f}|_k(x) = \widetilde{f}^{pie\_tan}|_k(x)$
11:     **else if** $f|_k'' \leq 0$ **then**
12:         $\overline{f}|_k(x) = \widetilde{f}^{pie\_tan}|_k(x)$
13:         $\underline{f}|_k(x) = \widetilde{f}^{lin\_int}|_k(x)$
14:     **end if**
15: **end for**

---

**Transformation into ReLU-Equivalent Form:** The approximating piecewise linear upper-lower functions are converted into a form that mimics the ReLU function's behavior; i.e., $\tilde{f}(x) = W^{(2)}\text{ReLU}(W^{(1)}x + b^{(1)}) + b^{(2)}$. This involves creating new weighting coefficients and intercepts that replicate the ReLU's activation pattern across the approximation

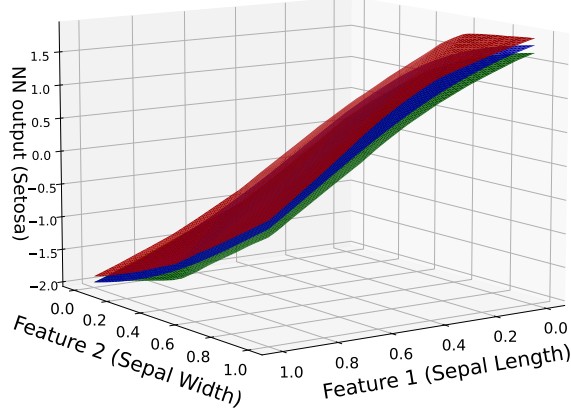

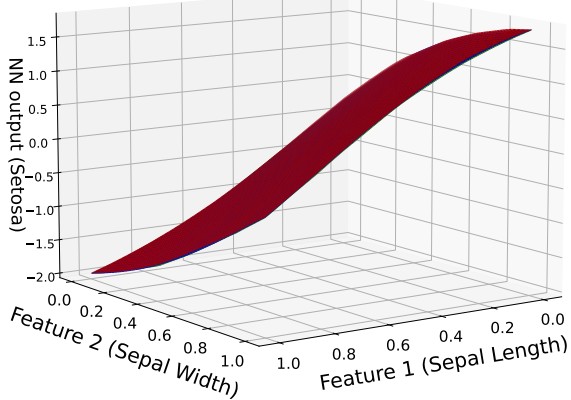

**Figure 2:** *Tanh* NN output for the *Setosa* class (blue) in the Iris dataset, and its ReLU NN upper (red) and lower (green) approximations with 5 (upper panel) and 10 (lower panel) segments of bounding per convex section.

intervals.

Specifically, we receive a set of intervals $\{[x_{i-1}, x_i]\}_{i=1}^n$ with the corresponding set of parameters of affine transformations $\{(v_i, c_i)\}_{i=1}^n$:

$$\widetilde{f}(\tau) = c_i + v_i\tau, \qquad \tau \in [x_{i-1}, x_i],$$

and set $v_0 = 0$. Then, the corresponding ReLU-equivalent definition of approximation $\widetilde{f}$ on $[x_0, x_n]$ is

$$\widetilde{f}(\tau) = x_0 v_1 + c_1 + \sum_{i=1}^n \xi_i \text{ReLU}(|v_i - v_{i-1}|(\tau - x_{i-1})),$$

$$\xi_i = \text{sign}(v_i - v_{i-1}).$$

**Full Neural Network Approximation:** The entire NN is approximated by applying the above techniques to each neuron, layer by layer, and then merging all intermediate weights and biases. For each neuron, both upper and lower approximations are generated, capturing the range of possible outputs under different inputs. To ensure the correct

propagation of approximators, to create an upper approximation, we connect the upper approximation of the external layer with the upper approximation of the internal subnetwork if the internal subnetwork has a positive coefficient, or with the lower approximation if it has a negative coefficient. The reverse applies to the lower approximation. Detailed explanation is provided in Lemma B.11 and B.12 in Appendix B.1. This leads to a composition of uniformly convergent sequences, guaranteeing the overall uniform convergence of the final estimator to the original neural network, as we show in Theorem 3.14. The final output is a set of piecewise linear approximations that bound the output of the original neural network, which can then be used for further analysis or verification.

*Example* 3.13. Using the same setup as in Example 3.10, we train a fully connected neural network with 3 layers of 12 neurons each with tanh activation followed by 1 layer of 3 output neurons with linear activation on the Iris dataset, focusing on the re-scaled features *Sepal Length* and *Sepal Width*. We construct upper and lower approximations of the network's output by ReLU neural networks with linear output layer. Two approximations are performed: with 5 and 10 bounding segments per convex section at each node (upper and lower panels of Figure 2, respectively). Notably, with 10 segments, the original network and its approximations are nearly indistinguishable.

This procedure applies to various non-monotonic functions, such as monomials $x^n, n \in \mathbb{N}$. These functions can be attained by a sequence of transformations that ensure monotonicity at all intermediate steps. These transformations can be represented as a subnetwork. Furthermore, multivariate functions like softmax and the product operation also have equivalent subnetworks with continuous monotonic transformations, as shown in Appendix C.

The main result of this section is Theorem 3.14 that shows that the upper and lower ReLU bounds converge uniformly and monotonically to the target NN.

**Theorem 3.14** (Uniform Monotonic Convergence of the ReLU Bounds). *The sequences of estimating functions that are ReLU NNs generated by the method in Section 3.2, establish upper and lower bounds for the target NN. These sequences are monotonically decreasing for the upper bounds and monotonically increasing for the lower bounds, and converge uniformly to the target network.*

The proof is given in Appendix B.2. It involves multiple steps, each outlined in different lemmas in Appendix B.1.

*Remark* 3.15. Theorem 3.11 follows from the UAT (Hornik et al., 1989). Theorem 3.14 is a special case of Theorem 3.11 but our proof is constructive and explicitly outlines the sequences that establish the bounds.

### 3.3. Application to an arbitrary function on a compact domain

We present the *universal distribution approximation theorem*, which may serve as a starting point for further research in the field of stochastic behavior of functions and the NNs that describe them.

**Theorem 3.16** (Universal distribution approximation theorem). *Let $\mathbf{X}$ be a random vector with continuous pdf $\phi(\mathbf{x})$ supported over a compact hyperrectangle $K \subset \mathbb{R}^{n_0}$. Let $Y = \mathcal{W}(\mathbf{x}) \in \mathbb{R}$ be a continuous function of $\mathbf{X}$ with domain $K$, and let $F(y)$ denote its cdf. Then, there exist sequences of cdf bounds $\{\underline{F}_n\}$, $\{\overline{F}_n\}$, $n = 1, 2, \ldots$, which can be constructed by bounding the distributions of sequences of ReLU NNs and such that*

$$\underline{F}_n(y) \leq F(y) \leq \overline{F}_n(y)$$

*for all $y \in \{\mathcal{W}(\mathbf{x}) : \mathbf{x} \in K\}$ and*

$$\underline{F}_n(y) \to F(y), \quad \overline{F}_n(y) \to F(y)$$

*for all $y$ where $F(y)$ is continuous. Moreover, if $\mathcal{W}(\mathbf{X})$ is almost surely nowhere locally constant, that is*

$$\int\limits_{\{\mathcal{W}(\mathbf{x}) = y\}} \phi(\mathbf{x}) d\mathbf{x} = 0 \tag{8}$$

*for all $y \in \{\mathcal{W}(\mathbf{x}) : \mathbf{x} \in K\}$, then both bounds $\underline{F}_n, \overline{F}_n$ converge uniformly to the true cdf $F$.*

The proof is presented in Appendix A.3. It leverages the UAT (Hornik et al., 1989) to approximate the function $\mathcal{W}(\mathbf{x})$ and input pdf $\phi(\mathbf{x})$ with ReLU NNs to arbitrary accuracy. The cdf bounds are then computed over polytope intersections. Increasing the NN complexity results in more simplices and thus leads to finer local affine approximations of the pdf.

To bound the cdf of a given NN with respect to a specified input pdf, we construct upper and lower bounding ReLU NNs to approximate the target NN. Next, we subdivide the resulting polytopes of the bounding ReLU NNs into simplices as much as needed to achieve the desired accuracy and locally bound the input pdf with constant values (the simplest polynomial form) on these simplices, transforming the problem into the one described in Section 3.1.

## 4. Experiments

We consider four datasets, Banana, Diabetes (Efron et al., 2004), Iris (Fisher, 1936), and Wine (Aeberhard & Forina, 1992)[6].

---

[6]Iris, Wine and Diabetes are provided in the Python package `scikit-learn` and Banana in Kaggle.

In all experiments, we compare our guaranteed bounds for the output cdf with cdf estimates obtained via a Monte Carlo (MC) simulation (100 million samples), and with the "Piecewise Linear Transformation" (PLT) method of Krapf et al. (2024) following their default subdivision setup. For each simplex an edgewise subdivision is performed with the number of subdivisions depending on the input dimension $n_0$. For input dimension $n_0$ from 1 to 3, the number of subdivisions was set to 250, 100 and 30, respectively. In contrast, our implementation uses a bound (arbitrarily set to $50\,000$) on the total number of vertices based on the iteratively refined triangulation.

Since the true cdf is contained within the limits computed by our method, the experiments assess the tightness of our bounds compared to both MC and PLT by tallying the number of "out of bounds" instances. Essentially, achieving very tight bounds makes it challenging to stay within those limits, whereas even imprecise estimates can fall within broader bounds.

Our experimental setup is based on small pre-trained (fixed) neural networks. For the Banana data, we trained a ReLU network with 2 hidden layers of 16 neurons each. For the Diabetes dataset, we trained a ReLU network with 3 hidden layers with 32, 16, and 8 neurons, respectively. The univariate output has no activation. Training was performed on 70% of the data, which were randomly selected. The remaining 30% comprise the test set. We added normal noise to the 1 or 2 randomly selected features (see $n_0$ in Table 1) of every observation with variance the sample variance of the selected feature(s) in the test set (different features in different observations). This is a simplified simulation setting of that of (Krapf et al., 2024).

For both Iris and Wine, we replicated the exact experimental setup from (Krapf et al., 2024) using the same test sets, Gaussian mixtures as randomness as well as the same pre-trained networks[7]. The 1 to 3 dimensional Gaussian mixtures were computed by first deleting 25% or 50% of the test data. Subsequently, 50 new observations are imputed using MICE (van Buuren & Groothuis-Oudshoorn, 2011). The new imputed observations where used as the dataset for a Gaussian kernel density estimate providing the Gaussian mixture for each of the original deleted observations. The only difference in the experimental setup is the MC estimate, which we recomputed with a higher sample count ($10^8$ as opposed to $8 \cdot 10^4$ till $1.024 \cdot 10^7$ with a median of $6.4 \cdot 10^5$ deduced by an incremental doubled sample size convergence criteria). Here, the "ground truth" are the exact cdf bounds our method computes as opposed to the MC estimate in the (Krapf et al., 2024) experiments, which was

---

[7]GitHub page `https://github.com/URWI2/Piecewise-Linear-Transformation`, accessed Jan 25, 2025

an approximation to the true cdf.

We summarize our results in Table 1. When the input dimension is 1, our bounds are very tight and we thus observe a high ratio of "out of bounds" samples for both MC and PLT compared to the number of grid points the cdf was evaluated at. This has two components: (a) In regions where the cdf is very flat, we obtain very tight bounds leading to small errors in a bucketed estimation approach easily falling outside of these tight bounds; (b) due to either pure random effect in the case of MC or numerical estimation inaccuracies in case of PLT, MC and PLT estimates are outside the bounds. Although the PLT estimator directly targets the pdf and would be expected to achieve greater precision than ours (as we target bounding the cdf instead), we note that in these examples, especially as regards PLT, a "coarse grid" can cause inaccuracies in areas where the pdf fluctuates significantly. Nevertheless, this behavior diminishes as the input dimension increases. Due to our hard bound on the maximum number of vertices in this simulation (namely 50 000), our estimated cdf bounds are wider in higher dimensions as a consequence of the curse of dimensionality.

## 5. Related Work

The literature on NN verification is not directly related to ours as it has been devoted to standard non-stochastic input NNs, where the focus is on establishing guarantees of local robustness. This line of work develops testing algorithms for whether the output of a NN stays the same within a specified neighborhood of the deterministic input (see, e.g., Gowal et al. (2019); Xu et al. (2020); Zhang et al. (2018; 2020); Shi et al. (2025); Bunel et al. (2020); Ferrari et al. (2022); Katz et al. (2017; 2019); Wu et al. (2024)).

To handle noisy data or aleatoric uncertainty (random input) in NNs, two main approaches have been proposed: sampling-based and probability density function (pdf) approximation-based. Sampling-based methods use Monte Carlo simulations to propagate random samples through the NN (see, e.g., Abdelaziz et al. (2015); Ji et al. (2020)), but the required replications to achieve similar accuracy to theoretical approaches such as ours, as can be seen in Table 1, can be massive. Pdf approximation-based methods assume specific distributions for inputs or hidden layers, such as Gaussian (Abdelaziz et al., 2015) or Gaussian Mixture Models (Zhang & Shin, 2021), but these methods often suffer from significant approximation errors and fail to accurately quantify predictive uncertainty. Comprehensive summaries and reviews of these approaches can be found in sources like Sicking et al. (2022) and Gawlikowski et al. (2023).

In the context of verifying neural network properties within a probabilistic framework, (Weng et al., 2019) proposed PROVEN, a general probabilistic framework that ensures robustness certificates for neural networks under Gaussian and Sub-Gaussian input perturbations with bounded support with a given probability. It employs CROWN (Zhang et al., 2018; 2020) to compute deterministic affine bounds and subsequently leverages straightforward probabilistic techniques based on Hoeffding's inequality (Hoeffding, 1963). PROVEN provides a probabilistically sound solution to ensuring the output of a NN is the same for small input perturbations with a given probability, its effectiveness hinges on the activation functions used. It cannot refine bounds or handle various input distributions, which may limit its ability to capture all adversarial attacks or perturbations in practical scenarios.

The most relevant published work to ours we could find in the literature is Krapf et al. (2024). They propagate input densities through NNs with piecewise linear activations like ReLU, without needing sampling or specific assumptions beyond bounded support. Their method calculates the propagated pdf in the output space using the piecewise linearity of the ReLU activation. (Krapf et al., 2024) estimate the output pdf and show experimentally that it is very close to the Monte Carlo based pdf. Despite its originality, the approach has drawbacks, as they compare histograms rather than the actual pdfs in their experiments. Theorem 5 (App. C) in (Krapf et al., 2024) suggests approximating the distribution with fine bin grids and input subdivisions, but this is difficult to implement in practice. Without knowledge of the actual distribution, it is challenging to define a sufficiently "fine" grid. In contrast, we compute exact bounds of the true output cdf over its entire support (at any point, no grid required) that depicts the maximum error over its support, and show convergence to the true cdf. Krapf et al. (2024) use a piecewise constant approximation for input pdfs, which they motivate by their Lemma 3 (App. C) to deduce that exact propagation of piecewise polynomials through a neural network is infeasible. We show that, in effect, it is possible and provide a method for exact integration over polytopes. Additionally, their approach is limited to networks with piecewise linear activations, excluding locally nonlinear functions. In contrast, our method adapts to NNs with continuous, monotonic piecewise *twice continuously differentiable* activations.

## 6. Conclusion

We develop a novel method to analyze the probabilistic behavior of the output of a neural network subject to noisy (stochastic) inputs. We formulate an algorithm to compute bounds (upper and lower) for the cdf of a neural network's output and prove that the bounds are guaranteed and that they converge uniformly to the true cdf.

Our approach enhances deterministic local robustness verification using non-random function approximation. By

**Table 1:** Comparison of our approach U/L-Dist (guaranteed upper and lower bounds) with pointwise estimators from Monte-Carlo simulations and PLT (Krapf et al., 2024). Under column headed by $n_0$ are the numbers of input variables, *# Tests* is test set size grouped by $n_0$, under *U/L-dist* are the mean distance (standard deviation) between our upper and lower bounds, under $\text{OOB}_\text{MC}$ and $\text{OOB}_\text{PLT}$ are the minimum, median and maximum number of points outside our bounds relative to *Grid Size* many cdf evaluation points for Monte-Carlo simulations and PLT, respectively. *Runtime* gives the mean computation time for PLT and our approach in seconds (MC took about 20 sec in all cases).

| Dataset | $n_0$ | # Tests | U/L-Dist | # Out of Bounds | | Grid | Runtime [sec] | |
| --- | --- | --- | --- | --- | --- | --- | --- | --- |
| | | | | MC | PLT | Size | PLT | OUR |
| | | | Mean (Std) | (Min/Median/Max) | (Min/Median/Max) | | Mean | Mean |
| banana | 1 | 1591 | 0.0001 (0.0000) | 0 / 103 / 896 | 77 / 565 / 999 | 1000 | 1.4 | 1.0 |
| diabetes | 1 | 133 | 0.0001 (0.0000) | 2 / 265 / 815 | 107 / 678 / 993 | 1000 | 1.2 | 1.0 |
| diabetes | 2 | 133 | 0.0015 (0.0009) | 0 / 1 / 92 | 0 / 207 / 992 | 1000 | 504 | 4.0 |
| $\text{iris}_{25}$ | 1 | 25 | 0.0000 (0.0000) | 44 / 282 / 868 | 1510 / 2245 / 2594 | 8000 | 0.23 | 5.9 |
| $\text{iris}_{25}$ | 2 | 10 | 0.0059 (0.0040) | 0 / 9 / 127 | 0 / 0 / 74 | 8000 | 38 | 20 |
| $\text{iris}_{25}$ | 3 | — | | | | | | |
| $\text{iris}_{50}$ | 1 | 20 | 0.0000 (0.0000) | 115 / 321 / 828 | 1794 / 2311 / 2976 | 8000 | 0.41 | 5.9 |
| $\text{iris}_{50}$ | 2 | 23 | 0.0064 (0.0064) | 0 / 37 / 413 | 0 / 9 / 119 | 8000 | 43 | 20 |
| $\text{iris}_{50}$ | 3 | 8 | 0.2019 (0.0414) | 0 / 18 / 128 | 0 / 31 / 178 | 8000 | 270 | 100 |
| $\text{wine2}_{25}$ | 1 | 32 | 0.0000 (0.0000) | 23 / 329 / 853 | 1864 / 2294 / 3026 | 8000 | 0.36 | 5.8 |
| $\text{wine2}_{25}$ | 2 | 8 | 0.0043 (0.0016) | 11 / 34 / 341 | 0 / 4 / 22 | 8000 | 44 | 19 |
| $\text{wine2}_{25}$ | 3 | 2 | 0.1777 (0.0068) | 300 / 449 / 598 | 60 / 63 / 66 | 8000 | 340 | 99 |
| $\text{wine2}_{50}$ | 1 | 27 | 0.0000 (0.0000) | 169 / 491 / 2129 | 1892 / 2191 / 2955 | 8000 | 0.41 | 5.9 |
| $\text{wine2}_{50}$ | 2 | 21 | 0.0053 (0.0028) | 0 / 70 / 615 | 0 / 4 / 46 | 8000 | 42 | 20 |
| $\text{wine2}_{50}$ | 3 | 13 | 0.1859 (0.0560) | 13 / 110 / 489 | 1 / 39 / 93 | 8000 | 300 | 99 |

bounding intermediate neurons with piecewise affine transformations and known ranges of activation functions evaluated with IBP (Gowal et al., 2019), we achieve more precise functional bounds. These bounds converge to the true functions of input variables as local linear units increase.

Our method targets neural networks with continuous monotonic piecewise twice continuously differentiable activation functions using tools like Marabou (Wu et al., 2024), originally designed for piecewise linear functions. While the current approach analyzes the behavior of NNs on a compact hyperrectangle, we can easily extend our theory to unions of bounded polytopes. In future research, we plan to bound the cdf of a neural network where the input admits arbitrary distributions with bounded piecewise continuous pdf supported on arbitrary compact sets. Moreover, we intend to improve the algorithmic performance so that our method applies to larger networks.

## Impact Statement

This paper presents work whose goal is to advance the field of Machine Learning. There are many potential societal consequences of our work, none which we feel must be specifically highlighted here.

## Acknowledgments

We would like to thank three reviewers for their helpful feedback and suggestions, which improved our work.

The research in this paper has been partially funded by the Vienna Science and Technology Fund (WWTF) grant [10.47379/ICT19018] (ProbInG), the TU Wien Doctoral College (SecInt), the FWF research project P 30690-N35, and WWTF project ICT22-023 (TAIGER).

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

# A. Proof of Theorems

## A.1. Theorem 3.8

Suppose the activation function in the prediction NN (2) is ReLU and $n_0$ and $n_L$ are the number of input and output neurons, respectively. The integral of a function over a given domain can be expressed as the sum of integrals over a partition of the domain (disjoint subdomains whose union constitutes the original domain). To compute the cdf of $\widetilde{\mathbf{Y}} = f_L(\mathbf{x})$ at $\mathbf{y}$, $F_{\widetilde{\mathbf{Y}}}(y) = \Pr[f_L(\mathbf{x}) \leq \mathbf{y}]$, we compute the sum of $\mathbb{P}[NN^j(\mathbf{x}) \leq y \mid \mathbf{x} \in \mathcal{P}_j]$, each of which is the integral of the pdf of the input over the given polytopes subject to $NN^j(\mathbf{x}) \leq \mathbf{y}$. These sets of polytopes $\{\mathcal{P}_j\}$ and the corresponding local affine transformations $\{NN^j(\mathbf{x})\}$ always exist, as shown in (Raghu et al., 2017).

$\mathcal{P}_j$ is a convex polytope and can be represented as the intersection of halfspaces (see (Ziegler, 1995)). The set $\{\mathbf{x} : NN^j(\mathbf{x}) = \mathbf{c}^j + \mathbf{V}^j\mathbf{x} \leq \mathbf{y}\}$ is defined as the intersection of halfspaces

$$\bigcap_{t=1}^{n_L} \left\{ \mathbf{x} : NN_t^j(\mathbf{x}) = c_t^j + \sum_{z=1}^{n_0} x_z v_{t,z}^j \leq y_t \right\},$$

which when intersected with $\mathcal{P}_j$ and $k_i$ defines the reduced complex polytope $\mathcal{P}_{j,i}^r$. The desired local probability, $\mathbb{P}[NN^j(\mathbf{x}) \leq \mathbf{y} \mid \mathbf{x} \in \mathcal{P}_j]$, is the integral $\mathcal{I}\left[\phi_i(\mathbf{x}); \mathcal{P}_{j,i}^r\right]$ of the pdf $\phi_i(\mathbf{x})$ over the reduced polytope $\mathcal{P}_{j,i}^r$.

Using the Delaunay triangulation (Delaunay, 1934) one can decompose any convex polytope $\mathcal{P}_{j,i}^r$ into a disjoint set of simplices $\mathcal{T}_{i,j,s}$. This triangulation allows us to compute the integral over the polytope as a sum of integrals over each simplex. Assuming that the pdf of the input is a piecewise polynomial allows us to use the algorithm from (Lasserre, 2021) to compute exact integrals over all simplices. The sum of all these localized integrals (probabilities) is the exact cdf value at point $\mathbf{y}$.

## A.2. Theorem 3.11

Since $\widetilde{Y}$ is a feedforward NN with continuous activation functions on a compact support $K \subset \mathbb{R}^{n_0}$, for any $\epsilon > 0$, let $\{\epsilon_n\}$, $\epsilon > \epsilon_n > 0$ be a decreasing sequence. By the UAT (Hornik et al., 1989), there exists a sequence of ReLU networks $\{Y_{\epsilon_n}\}$, such that

$$\sup_K \|\widetilde{Y}(\mathbf{x}) - Y_{\epsilon_n}(\mathbf{x})\|_{\mathbb{R}^{n_0}} < \epsilon_n.$$

Setting $\underline{Y}'_n(\mathbf{x}) = Y_{\epsilon_n}(\mathbf{x}) - \epsilon_n$ and $\overline{Y}'_n(\mathbf{x}) = Y_{\epsilon_n}(\mathbf{x}) + \epsilon_n$, we have

$$\underline{Y}'_n(\mathbf{x}) \leq \widetilde{Y}(\mathbf{x}) \leq \overline{Y}'_n(\mathbf{x}).$$

Now, we let $\underline{Y}_n(\mathbf{x}) = \max_{1 \leq i \leq n} \underline{Y}'_n(\mathbf{x})$, which is still not greater than $\widetilde{Y}(\mathbf{x})$, and $\overline{Y}_n(\mathbf{x}) = \min_{1 \leq i \leq n} \overline{Y}'_n(\mathbf{x})$, which is still not smaller than $\widetilde{Y}(\mathbf{x})$. One can see that $\{\overline{Y}_n\}$ and $\{\underline{Y}_n\}$ are monotonically decreasing and increasing, respectively. It should be noted that the $\min$ and $\max$ operators can be represented as ReLU networks (see Fig. 4), and the composition of ReLU networks is itself a ReLU network.

## A.3. Theorem 3.16

According to the UAT (Hornik et al., 1989), for any $\epsilon > 0$ there exist one-layer networks $\widetilde{Y}, \widetilde{\phi}$ with ReLU activation function, such that

$$\sup_{\mathbf{x} \in K} \| \mathcal{W}(\mathbf{x}) - \widetilde{Y}(\mathbf{x}) \| < \epsilon, \qquad \sup_{\mathbf{x} \in K} \| \phi(\mathbf{x}) - \widetilde{\phi}(\mathbf{x}) \| < \epsilon.$$

Define $\underline{Y}_n(\mathbf{x}) = \widetilde{Y}(\mathbf{x}) - \epsilon$, which is also a NN. Similarly, for $\overline{Y}_n, \underline{\phi}_n, \overline{\phi}_n$. Then,

$$\mathcal{W}(\mathbf{x}) - 2\epsilon \leq \underline{Y}_n(\mathbf{x}) = \widetilde{Y}(\mathbf{x}) - \epsilon \leq \mathcal{W}(\mathbf{x}) \leq \widetilde{Y}(\mathbf{x}) + \epsilon = \overline{Y}_n(\mathbf{x}) < \mathcal{W}(\mathbf{x}) + \epsilon$$

$$\phi(\mathbf{x}) - 2\epsilon < \underline{\phi}_n(\mathbf{x}) = \widetilde{\phi}(\mathbf{x}) - \epsilon \leq \phi(\mathbf{x}) \leq \widetilde{\phi}(\mathbf{x}) + \epsilon = \overline{\phi}_n(\mathbf{x}) < \phi(\mathbf{x}) + 2\epsilon,$$

Letting $\epsilon \to 0$ results in $\underline{Y}_n, \overline{Y}_n \to \mathcal{W}$ and $\underline{\phi}_n, \overline{\phi}_n \to \phi$, uniformly on a compact domain while guaranteeing that they be lower/upper bounds.

Let

$$\overline{F}_n(y) = \min\left[1, \int_{\{\mathbf{x}:\mathbf{x}\in K\cap \underline{Y}_n(\mathbf{x})\leq y\}} \overline{\phi}_n(\mathbf{x})d\mathbf{x}\right], \quad \underline{F}_n(y) = \max\left[0, \int_{\{\mathbf{x}:\mathbf{x}\in K\cap \overline{Y}_n(\mathbf{x})\leq y\}} \underline{\phi}_n(\mathbf{x})d\mathbf{x}\right]$$

The limit cdf is

$$F(y) = \int_{\{\mathbf{x}:\mathbf{x}\in K\cap \mathcal{W}(\mathbf{x})\leq y\}} \phi(\mathbf{x})d\mathbf{x}$$

Since $\underline{\phi}_n(\mathbf{x}) \leq \phi(\mathbf{x}) \leq \overline{\phi}_n(\mathbf{x})$ and $\underline{Y}_n(\mathbf{x}) \leq \mathcal{W}(\mathbf{x}) \leq \overline{Y}_n(\mathbf{x})$ for any $\mathbf{x} \in K$, $\{\mathbf{x} : \mathbf{x} \in K \cap \underline{Y}_n(\mathbf{x}) \leq y\} \supseteq \{\mathbf{x} : \mathbf{x} \in K \cap \mathcal{W}(\mathbf{x}) \leq y\}$ and $\{\mathbf{x} : \mathbf{x} \in K \cap \overline{Y}_n(\mathbf{x}) \leq y\} \subseteq \{\mathbf{x} : \mathbf{x} \in K \cap \mathcal{W}(\mathbf{x}) \leq y\}$. Since $0 \leq F(y) \leq 1$ for all $y$,

$$\underline{F}_n(y) \leq F(y) \leq \overline{F}_n(y)$$

for all $y \in \{\mathcal{W}(\mathbf{x}) : \mathbf{x} \in K\}$.

Now let us fix an arbitrary $y = \mathcal{W}(\mathbf{x})$ for $\mathbf{x} \in K$, such that $y$ is a continuity point of $F$.

$$\overline{F}_n(y) - F(y) \leq \int_{\{\mathbf{x}:\mathbf{x}\in K\cap\underline{Y}_n(\mathbf{x})\leq y\}} \overline{\phi}_n(\mathbf{x})dx - \int_{\{\mathbf{x}:\mathbf{x}\in K\cap\mathcal{W}(\mathbf{x})\leq y\}} \phi(\mathbf{x})d\mathbf{x}$$

$$= \underbrace{\int_{\{\mathbf{x}:\mathbf{x}\in K\cap\underline{Y}_n(\mathbf{x})\leq y\}} (\overline{\phi}_n(\mathbf{x}) - \phi(\mathbf{x}))dx}_{A} + \underbrace{\int_{\{\mathbf{x}:\mathbf{x}\in K\cap\underline{Y}_n(\mathbf{x})\leq y\}} \phi(\mathbf{x})dx - \int_{\{\mathbf{x}:\mathbf{x}\in K\cap\mathcal{W}(\mathbf{x})\leq y\}} \phi(\mathbf{x})dx}_{B}$$

$$F(y) - \underline{F}_n(y) \leq \int_{\{\mathbf{x}:\mathbf{x}\in K\cap\mathcal{W}(\mathbf{x})\leq y\}} \phi(\mathbf{x})dx - \int_{\{\mathbf{x}:\mathbf{x}\in K\cap\overline{Y}_n(x)\leq y\}} \underline{\phi}_n(\mathbf{x})d\mathbf{x}$$

$$= \underbrace{\int_{\{\mathbf{x}:\mathbf{x}\in K\cap\overline{Y}_n(\mathbf{x})\leq y\}} (\phi(\mathbf{x}) - \underline{\phi}_n(\mathbf{x}))dx}_{C} + \underbrace{\int_{\{\mathbf{x}:\mathbf{x}\in K\cap\mathcal{W}(\mathbf{x})\leq y\}} \phi(\mathbf{x})d\mathbf{x} - \int_{\{\mathbf{x}:\mathbf{x}\in K\cap\overline{Y}_n(\mathbf{x})\leq y\}} \phi(\mathbf{x})d\mathbf{x}}_{D}$$

The left integrals in both equations (A and C) converge to zero due to the uniform convergence to zero of the integrands over the whole set $K$. The second differences (B and D) converge to zero, since the superset $\{\mathbf{x} : \mathbf{x} \in K \cap \underline{Y}_n(\mathbf{x}) \leq y\}$ and subset $\{\mathbf{x} : \mathbf{x} \in K \cap \overline{Y}_n(\mathbf{x}) \leq y\}$ of the limits of integrals, respectively, converge to the true limit set $\{\mathbf{x} : \mathbf{x} \in K \cap \mathcal{W}(\mathbf{x}) \leq y\}$ due to continuity. We have proven pointwise convergence for every point of continuity of the limiting cdf.

Requiring 8 means that the limiting distribution has no point mass; i.e., it is continuous. The support of $Y$ is compact because it is the continuous image of the compact set $K$. We can then apply Polya's theorem (Rao, 1962) that the convergence of both bounds is uniform as sequences of monotonically increasing functions converging pointwise to a continuous function on a compact set.

## B. Convergence of the ReLU bounds

### B.1. Preliminary results

**Lemma B.1.** *Let $a_k, a_{k+1} \in \mathbb{R}$ with $a_{k+1} > a_k$ and assume $f : [a_k, a_{k+1}] \to \mathbb{R}$ is twice continuously differentiable, monotone increasing and strictly convex (concave) on $[a_k, a_{k+1}]$. Then the values of the linear interpolation and piecewise tangent approximation at boundary points coincide with the original function. That is, $\widetilde{f}^{lin\_int}(a_k) = \widetilde{f}^{pie\_tan}(a_k) = f(a_k)$ and $\widetilde{f}^{lin\_int}(a_{k+1}) = \widetilde{f}^{pie\_tan}(a_{k+1}) = f(a_{k+1})$.*

*Proof.*

$$\widetilde{f}^{lin\_int}(a_k) = f(a_k) + (a_k - a_k)\kappa_1 = f(a_k),$$

$$\begin{aligned}
\widetilde{f}^{lin\_int}(a_{k+1}) &= f(a_{k'}^{lin\_int}) + (a_{k+1} - a_{k'}^{lin\_int})\kappa_2 \\
&= f(a_{k'}^{lin\_int}) + (a_{k+1} - a_{k'}^{lin\_int})\frac{f(a_{k+1}) - f(a_{k'}^{lin\_int})}{a_{k+1} - a_{k'}^{lin\_int}} \\
&= f(a_{k+1}),
\end{aligned}$$

$$\widetilde{f}^{pie\_tan}(a_k) = f(a_k) + f_+'(a_k)(a_k - a_k) = f(a_k),$$

$$\widetilde{f}^{pie\_tan}(a_{k+1}) = f(a_{k+1}) + f_-'(a_{k+1})(a_{k+1} - a_{k+1}) = f(a_{k+1}).$$

$\square$

**Lemma B.2.** *Let $[a_k, a_{k+1}] \in \mathbb{R}$ be a closed interval with $a_{k+1} > a_k$, and let $f : [a_k, a_{k+1}] \to \mathbb{R}$ be twice continuously differentiable, monotonically increasing, and strictly convex (or concave) on $[a_k, a_{k+1}]$. Then, the intermediate points lie strictly within the interval $[a_k, a_{k+1}]$, $a_k < a_{k'}^{lin\_int} < a_{k+1}$ and $a_k < a_{k'}^{pie\_tan} < a_{k+1}$, and the functions of linear interpolation and piecewise tangent approximation are continuous on $[a_k, a_{k+1}]$.*

*Proof.* For linear interpolation, the statement follows immediately from the definition. Specifically, for the midpoint interpolation, we have

$$a_{k'}^{lin\_int} = \frac{a_k + a_{k+1}}{2},$$

$$a_k < \frac{a_k + a_{k+1}}{2} < a_{k+1}.$$

For piecewise tangent approximation, we define $a_{k'}^{pie\_tan}$ as

$$a_{k'}^{pie\_tan} = \frac{f(a_k) - f(a_{k+1}) - \big(f_+'(a_k)a_k - f_-'(a_{k+1})a_{k+1}\big)}{f_-'(a_{k+1}) - f_+'(a_k)}.$$

Consider first the case where $f$ is convex on $[a_k, a_{k+1}]$. By the convexity of $f$, we have the inequality

$$(a_k - a_{k+1})f_-'(a_{k+1}) < f(a_k) - f(a_{k+1}) < (a_k - a_{k+1})f_+'(a_k).$$

Adding the terms $f_-'(a_{k+1})a_{k+1} - f_+'(a_k)a_k$ to each part of the inequality, we get

$$\begin{aligned}
a_k f_-'(a_{k+1}) - a_k f_+'(a_k) &< f(a_k) - f(a_{k+1}) - f_+'(a_k)a_k + f_-'(a_{k+1})a_{k+1} \\
&< a_{k+1}f_-'(a_{k+1}) - a_{k+1}f_+'(a_k).
\end{aligned}$$

Since the denominator of $a_{k'}^{pie\_tan}$, i.e., $f_-'(a_{k+1}) - f_+'(a_k)$, is strictly positive by convexity, dividing the entire inequality by this denominator yields

$$a_k < a_{k'}^{pie\_tan} < a_{k+1}.$$

In the case where $f$ is concave on $[a_k, a_{k+1}]$, we have a similar inequality:

$$(a_{k+1} - a_k)f_-'(a_{k+1}) < f(a_{k+1}) - f(a_k) < (a_{k+1} - a_k)f_+'(a_k).$$

Adding the terms $f_+'(a_k)a_k - f_-'(a_{k+1})a_{k+1}$ to each part of the inequality, we get

$$\begin{aligned}
a_k f_+'(a_k) - a_k f_-'(a_{k+1}) &< f(a_{k+1}) - f(a_k) + f_+'(a_k)a_k - f_-'(a_{k+1})a_{k+1} \\
&< a_{k+1}f_+'(a_k) - a_{k+1}f_-'(a_{k+1}).
\end{aligned}$$

Since the denominator $f'_-(a_{k+1}) - f'_+(a_k)$ of $a_{k'}^{pie\_tan}$ is strictly negative for concave functions, dividing the entire inequality by this negative denominator yields

$$a_k < a_{k'}^{pie\_tan} < a_{k+1}.$$

Next, we show that the interpolation functions are continuous at the intermediate point. For linear interpolation, we check the continuity by verifying that the two parts meet at $a_{k'}^{lin\_int}$. We have

$$f(a_{k'}^{lin\_int}) = f(a_{k'}^{lin\_int}) + (a_{k'}^{lin\_int} - a_{k'}^{lin\_int})\kappa_2 \stackrel{\text{right}}{=} \widetilde{f}^{lin\_int}(a_{k'}^{lin\_int}),$$

$$\stackrel{\text{left}}{=} f(a_k) + (a_{k'}^{lin\_int} - a_k)\kappa_1 = f(a_k) + (a_{k'}^{lin\_int} - a_k)\frac{f(a_{k'}^{lin\_int}) - f(a_k)}{a_{k'}^{lin\_int} - a_k} = f(a_{k'}^{lin\_int}).$$

Thus, the two parts of the linear interpolation meet at $a_{k'}^{lin\_int}$, and $\widetilde{f}^{lin\_int}$ is continuous at $a_{k'}^{lin\_int}$.

For piecewise tangent approximation, we check that the left and right parts meet at $a_{k'}^{pie\_tan}$. From the left, we have

$$\widetilde{f}^{pie\_tan}(\tau) = f(a_k) + f'_+(a_k)(\tau - a_k), \quad \tau \in [a_k, a_{k'}^{pie\_tan}].$$

Substituting $\tau = a_{k'}$, we get

$$\widetilde{f}^{pie\_tan}(a_{k'}^-) = f(a_k) + f'_+(a_k)(a_{k'}^- - a_k).$$

From the right, we have

$$\widetilde{f}^{pie\_tan}(\tau) = f(a_{k+1}) + f'_-(a_{k+1})(\tau - a_{k+1}), \quad \tau \in [a_{k'}^{pie\_tan}, a_{k+1}].$$

Substituting $\tau = a_{k'}$, we get

$$\widetilde{f}^{pie\_tan}(a_{k'}^+) = f(a_{k+1}) + f'_-(a_{k+1})(a_{k'}^+ - a_{k+1}).$$

Setting these equal, we have

$$f(a_k) + f'_+(a_k)(a_{k'} - a_k) = f(a_{k+1}) + f'_-(a_{k+1})(a_{k'} - a_{k+1}).$$

Solving for $a_{k'}$, we obtain

$$a_{k'} = \frac{f(a_k) - f(a_{k+1}) - (a_k f'_+(a_k) - a_{k+1} f'_-(a_{k+1}))}{f'_-(a_{k+1}) - f'_+(a_k)} = a_{k'}^{pie\_tan}.$$

This confirms the continuity of $\widetilde{f}^{pie\_tan}(\tau)$ at $a_{k'}^{pie\_tan}$.

$\square$

**Lemma B.3.** *Let $a_{k+1} > a_k$ and $[a_k, a_{k+1}] \subset \mathbb{R}$, and let $f : [a_k, a_{k+1}] \to \mathbb{R}$ be a twice continuously differentiable, monotonically increasing, and strictly convex (or strictly concave) function on $[a_k, a_{k+1}]$. Then, the estimating functions defined by linear interpolation $\widetilde{f}^{lin\_int}$ and piecewise tangent approximation $\widetilde{f}^{pie\_tan}$ are non-decreasing on $[a_k, a_{k+1}]$.*

*Proof.* We prove that both local estimators are non-decreasing functions.

*Case 1: Linear Interpolation.* The slopes of the linear interpolation segments are given by

$$\kappa_1 = \frac{f(a_{k'}^{lin\_int}) - f(a_k)}{a_{k'}^{lin\_int} - a_k} > 0, \quad \kappa_2 = \frac{f(a_{k+1}) - f(a_{k'}^{lin\_int})}{a_{k+1} - a_{k'}^{lin\_int}} > 0.$$

Since the original function $f$ is non-decreasing and $a_k < a_{k'}^{lin\_int} < a_{k+1}$, as established by Lemma B.2, both slopes are non-negative, ensuring that the interpolated function is non-decreasing.

*Case 2: Piecewise Tangent Approximation.* The derivatives of both the left and right segments of the piecewise tangent approximation are non-negative due to the increasing nature of the approximated function. Furthermore, by Lemma B.2, we have $a_k < a_{k'}^{pie\_tan} < a_{k+1}$, and the approximation remains continuous everywhere. Since both segments are non-decreasing linear functions, their combination also results in a non-decreasing estimator over $[a_k, a_{k+1}]$.

Thus, both estimation methods preserve the monotonicity of $f$.

$\square$

**Lemma B.4.** *Let $[\underline{a}, \overline{a}] \subset \mathbb{R}$ be a closed interval, and let $f : [\underline{a}, \overline{a}] \to \mathbb{R}$ be a continuous function satisfying $f \in \mathcal{C}^2_{p.w.}([\underline{a}, \overline{a}])$. Assume that there exist points $\underline{a} = a_1 < a_2 < \cdots < a_{n+1} = \overline{a}$ for some $n \in \mathbb{N}$ such that $f$ is twice continuously differentiable, monotonically increasing, and either strictly convex, strictly concave, or linear on each subinterval $[a_k, a_{k+1}]$ for $1 \leq k \leq n$. Then, the approximation method described in Section 3.2 constructs a continuous, non-decreasing estimating function $\widetilde{f}(x)$ over the entire interval $[\underline{a}, \overline{a}]$.*

*Proof.* The claim follows from the following observations:

a) Each subinterval $[a_k, a_{k+1}]$ is suitable for approximating $f$ using either linear interpolation, piecewise tangent approximation, or local linear approximation.

b) All the methods mentioned in (a) ensure that the estimator matches the original function at the boundary points of each subinterval, by Lemma B.1.

c) The approximations described in (a) are continuous within their respective subintervals (see Lemma B.2).

d) The methods in (a) produce non-decreasing functions within each subinterval (see Lemma B.3).

By sequentially linking the estimators across all segments $\{[a_k, a_{k+1}]\}$, we obtain a continuous, non-decreasing, piecewise linear function $\widetilde{f}(x)$ over $[\underline{a}, \overline{a}]$. $\qquad\square$

**Lemma B.5** (Image of the local estimator). *Let $f : [\underline{a}, \overline{a}] \to \mathbb{R}$ be continuous, $f \in \mathcal{C}^2_{p.w.}([\underline{a}, \overline{a}])$ with $\underline{a} = a_1 < a_2 < \ldots < a_{n+1} = \overline{a}$ for $n \in \mathbb{N}$, such that $f\big|_{[a_k, a_{k+1}]}$ is twice continuously differentiable, monotonic increasing and either strictly convex (concave), or linear on $[a_k, a_{k+1}]$ for $1 \leq k \leq n$. Then the image of the estimating function $\widetilde{f}(x)$ defined by the approximation method in Section 3.2 coincides with the image of the target function $f$, that is $f([\underline{a}, \overline{a}]) = \widetilde{f}([\underline{a}, \overline{a}])$.*

*Proof.* Since function $f$ is continuous on a compact $[\underline{a}, \overline{a}]$, and monotonic, then it maps $[\underline{a}, \overline{a}]$ into the closed interval (compact) $[f(\underline{a}), f(\overline{a})] \in \mathbb{R}$. But the estimator $\widetilde{f}$ is also continuous monotonic function $[\underline{a}, \overline{a}]$, and by Lemmas B.1, B.2, $f(\underline{a}) = \widetilde{f}(\underline{a})$ and $f(\overline{a}) = \widetilde{f}(\overline{a})$. That is why, the ranges of values of $f$ and $\widetilde{f}$ on $[\underline{a}, \overline{a}]$ coincide and equal to $[f(\underline{a}), f(\overline{a})] \in \mathbb{R}$. $\qquad\square$

**Theorem B.6** (Convergence of piecewise linear interpolation (Driscoll & Braun, 2018), Ch.5). *Suppose that $f(x)$ has a continuous second derivative in $[a_k, a_{k+1}]$, that is $f \in C^2([a_k, a_{k+1}])$. Let $p_{n_{int}}(x)$ be the piecewise linear interpolant of $(a_{k_i}, f(a_{k_i}))$ for $i = 0, \ldots, n_{int}$, where*

$$a_{k_i} = a_k + ih, \quad h = \frac{a_{k+1} - a_k}{n_{int}}.$$

*Then, the error bound satisfies*

$$\|f - p_{n_{int}}\|_\infty = \max_{x \in [a_k, a_{k+1}]} |f(x) - p_{n_{int}}(x)| \leq Mh^2,$$

*where*

$$M = \max_{[a_k, a_{k+1}]} f''(x)$$

**Theorem B.7** (Convergence of piecewise tangent approximation). *Suppose that $f \in C^2([a_k, a_{k+1}])$ and is strictly convex (or concave) in $[a_k, a_{k+1}]$. Let $\widetilde{f}^{pie\_tan}_{n_{tan}}(x)$ be the piecewise tangent approximator over subsegments $\left[a_{k_i}, a_{k_{i+1}}\right]$ for $i = 0, \ldots, n_{tan}$, where*

$$a_{k_i} = a_k + ih, \quad h = \frac{a_{k+1} - a_k}{n_{tan}}.$$

*Then, the error bound satisfies*

$$\|f - \widetilde{f}^{pie\_tan}_{n_{tan}}\|_\infty = \max_{x \in [a_k, a_{k+1}]} |f(x) - \widetilde{f}^{pie\_tan}_{n_{tan}}(x)| \leq Mh^2,$$

*where*

$$M = \max_{[a_k, a_{k+1}]} f''(x)$$

*Proof.* Each element of the piecewise tangent approximation is the Taylor series expansion of the first order around the boundary point of the subsegment. We consider the double Taylor series approximation on the refinement $\left[a_{k_i}, a_{k_{i+1}}\right]$ of the segment $[a_k, a_{k+1}]$ for $i = 0, \dots, n_{int}$, where

$$a_{k_i} = a_k + ih, \quad h = \frac{a_{k+1} - a_k}{n_{tan}}.$$

Since for the twice continuously differentiable in $\left[a_{k_i}, a_{k_{i+1}}\right]$ the Lagrange Remainder of the Taylor series expansion (Rudin, 1976), which represents an error term, can be bounded with

$$\max_{\left[a_{k_i}, a_{k_i'}^{pie\_tan}\right]} \left[ f(x) - \widetilde{f}^{pie\_tan}(x) \right] \leq M_{i_1} \frac{(a_{k_i'}^{pie\_tan} - a_{k_i})^2}{2} \leq M_{i_1} \frac{(a_{k_{i+1}} - a_{k_i})^2}{2},$$

$$\max_{\left[a_{k_i'}^{pie\_tan}, a_{k_{i+1}}\right]} \left[ f(x) - \widetilde{f}^{pie\_tan}(x) \right] \leq M_{i_2} \frac{(a_{k_{i+1}} - a_{k_i'}^{pie\_tan})^2}{2} \leq M_{i_2} \frac{(a_{k_{i+1}} - a_{k_i})^2}{2},$$

where

$$M_{i_1} = \max_{\left[a_{k_i}, a_{k_{i+1}'}^{pie\_tan}\right]} f''(x) \leq \max_{[a_k, a_{k+1}]} f''(x) = M$$

$$M_{i_2} = \max_{\left[a_{k_i'}^{pie\_tan}, a_{k_{i+1}}\right]} f''(x) \leq \max_{[a_k, a_{k+1}]} f''(x) = M$$

The maximum $M$ exists and is attainable due to the continuity of the second derivative on a compact $[a_k, a_{k+1}]$. That is, the maximum error on the whole segment of approximation can be bounded as

$$M \left[ \frac{a_{k+1} - a_k}{n_{tan}} \right]^2 = M h^2 \xrightarrow[n_{tan} \to \infty]{} 0$$

$\square$

The following lemma demonstrates that the approximation procedure presented in Section 3.2 generates sequences of estimators that: i) serve as valid bounds for the target function, and ii) converge monotonically to the target function. This implies that each new estimator can only improve upon the previous one.

**Lemma B.8.** *Suppose that $f \in C^2([a_k, a_{k+1}])$ is monotonic increasing and strictly convex (or concave) on $[a_k, a_{k+1}]$. Let $\widetilde{f}_n^{lin\_int}(x)$ be the piecewise linear interpolant and $\widetilde{f}_n^{pie\_tan}(x)$ be the piecewise tangent approximator over subsegments $\left[a_{k_i}, a_{k_{i+1}}\right]$ for $i = 0, \dots, 2^n$, where*

$$a_{k_i} = a_k + ih, \quad h = \frac{a_{k+1} - a_k}{2^n},$$

*with $n \in \mathbb{N}$. Then the estimating functions defined by the linear interpolation $\widetilde{f}_n^{lin\_int}$ and piecewise tangent approximation $\widetilde{f}_n^{pie\_tan}$ define upper (lower) and lower (upper), respectively, bounds on the target function $f$. Moreover, $\{\widetilde{f}_n^{lin\_int}(x)\}_n$ and $\{\widetilde{f}_n^{pie\_tan}(x)\}_n$ are non-increasing (non-decreasing) and non-decreasing (non-increasing) sequences, respectively.*

*Proof.* By the definition of the convex (concave) function,

$$f(\alpha x_1 + (1 - \alpha)x_2) \leq (\geq) \alpha f(x_1) + (1 - \alpha) f(x_2)$$

for any $\alpha \in [0, 1]$ and $x_1, x_2$ from the region of convexity, and plot of the linear interpolant between any $x_1, x_2$ lies above (below) the plot of the function. That is, linear interpolation is always an upper (lower) approximation of the convex (concave) function.

On the other hand, any tangent line lies below (above) the plot of the convex (concave) function. Indeed, since on a convex segment the derivative of the function increases, that is $f'_+(a_{k_i}) \leq f'(x) \leq f'_-(a_{k_{i+1}})$ for all $x \in [a_{k_i}, a_{k_{i+1}}]$, then

$$f(x) = f(a_{k_i}) + \int_{a_{k_i}}^{x} f'(t)dt \geq f(a_{k_i}) + \int_{a_{k_i}}^{x} f'_+(a_{k_i})dt$$

$$= f(a_{k_i}) + f'_+(a_{k_i})(x - a_{k_i}), \quad x \in [a_{k_i}, a_{k'_i}]$$

$$f(x) = f(a_{k_{i+1}}) - \int_{a_{k_{i+1}}}^{x} f'(t)dt \geq f(a_{k_{i+1}}) - \int_{a_{k_{i+1}}}^{x} f'_-(a_{k_{i+1}})dt$$

$$= f(a_{k_{i+1}}) + f'_-(a_{k_{i+1}})(x - a_{k_{i+1}}), \quad x \in [a_{1'}, a_{k_{i+1}}]$$

Similarly, we can show that the piecewise tangent upper bounds the true concave function. Without loss of generality, we consider the case of a convex segment. We fix $n$. The current element of the sequences of piecewise linear interpolants $\widetilde{f}_n^{lin\_int}(x)$ includes the local linear item based on the interval $[t_1, t_2]$. The case of a concave segment is analogous. We define the current local linear approximation of the element of sequence of upper approximation as

$$\widetilde{f}_{\{t\}_n}^{lin\_int}(x) = f(t_1) + (x - t_1)\frac{f(t_2) - f(t_1)}{t_2 - t_1}$$

Let $\alpha$ be such that $0 \leq \alpha \leq 1$ and define a new point $t$ as a convex combination $t = \alpha t_1 + (1 - \alpha)t_2$. Let us show that $\widetilde{f}_{\{t\}_{n+1}}^{lin\_int}(x)(x) \leq \widetilde{f}_{\{t\}_n}^{lin\_int}(x)$ for $x \in [t_1, t_2]$, where

$$\widetilde{f}_{\{t\}_{n+1}}^{lin\_int}(x) = \begin{cases} f(t_1) + (x - t_1)\frac{f(t) - f(t_1)}{t - t_1}, & x \in [t_1, t] \\ \\ f(t) + (x - t)\frac{f(t_2) - f(t)}{t_2 - t}, & x \in [t, t_2] \end{cases}$$

Taking into account the convex segment,

$$f(t_1) + (x - t_1)\frac{f(t) - f(t_1)}{t - t_1} = f(t_1) + (x - t_1)\frac{f(\alpha t_1 + (1 - \alpha)t_2) - f(t_1)}{\alpha t_1 + (1 - \alpha)t_2 - t_1}$$

$$\leq f(t_1) + (x - t_1)\frac{\alpha f(t_1) + (1 - \alpha)f(t_2) - f(t_1)}{(1 - \alpha)(t_2 - t_1)}$$

$$= f(t_1) + (x - t_1)\frac{f(t_2) - f(t_1)}{t_2 - t_1}$$

$$f(t) + (x - t)\frac{f(t_2) - f(t)}{t_2 - t} = f(\alpha t_1 + (1 - \alpha)t_2)$$

$$+ (x - \alpha t_1 + (1 - \alpha)t_2)\frac{f(t_2) - f(\alpha t_1 + (1 - \alpha)t_2)}{t_2 - \alpha t_1 + (1 - \alpha)t_2}$$

$$\leq \alpha f(t_1) + (1 - \alpha)f(t_2)$$

$$+ (x - t_1)\frac{\alpha f(t_1) + (1 - \alpha)f(t_2) - f(t_2)}{\alpha t_1 + (1 - \alpha)t_2 - t_2}$$

$$+ (t_1 - \alpha t_1 + (1 - \alpha)t_2)\frac{\alpha f(t_1) + (1 - \alpha)f(t_2) - f(t_2)}{\alpha t_1 + (1 - \alpha)t_2 - t_2}$$

$$= f(t_1) + (x - t_1)\frac{f(t_2) - f(t_1)}{t_2 - t_1}$$

Since the refinement on each subsegment leads to the reduced next element of the sequence, the sequence is decreasing on the whole convex segment.

We next consider the piecewise tangent approximation on $[t_1, t_2]$,

$$\widetilde{f}^{pie\_tan}_{\{t\}_n}(x) = \begin{cases} f(t_1) + f'_+(t_1)(x - t_1), & x \in [t_1, t_{1'}], \\ f(a_2) + f'_-(t_2)(x - t_2), & x \in [t_{1'}, t_2], \end{cases}$$

where $t_{1'}$ is the point of intersection of the tangents. If we choose some parameter $\alpha$, where $0 \le \alpha \le 1$, and define the corresponding intermediate point as $t_* = \alpha t_1 + (1 - \alpha)t_2$, then the refined approximation is given by:

$$\widetilde{f}^{pie\_tan}_{\{t\}_{n+1}}(x) = \begin{cases} f(t_1) + f'_+(t_1)(x - t_1), & x \in [t_1, t_{1*}], \\ f(t_*) + f'(t_*)(x - t_*), & x \in [t_{1*}, t_{2*}], \\ f(t_2) + f'_-(t_2)(x - t_2), & x \in [t_{2*}, t_2], \end{cases}$$

where $t_{1*}$ is the intersection point of the left and middle lines, and $t_{2*}$ is the intersection point of the middle and right lines. We aim to show that $\widetilde{f}^{pie\_tan}_{\{t\}_{n+1}}(x)(x) \ge \widetilde{f}^{pie\_tan}_{\{t\}_n}(x)$ for $x \in [t_1, t_2]$.

Thus, the plot of the linear function corresponding to the middle curve lies above that of the left curve for $x > t_{1*}$ and above that of the right curve for $x < t_{2*}$ due to the monotonicity of the estimator, by Lemma B.3. Consequently, the refined estimator $\underline{f}_2$ coincides with the previous estimator $\underline{f}_1$ on the left and right segments, i.e., for $x \in [t_1, t_{1*}]$ and $x \in [t_{2*}, t_2]$, while it takes higher values for $x \in [t_{1*}, t_{2*}]$. To confirm this, it remains to show that $t_{1*} \le t_{1'} \le t_{2*}$.

First, we note that $t_1 \le t_{1'} \le t_2$, by Lemma B.2. This automatically leads to $t_1 \le t_{1*} \le t_* \le t_{2*} \le t_2$.

Consider the function

$$f_{\text{left}}(x) = \frac{x f'_-(x) - f(x) - C}{f'_-(x) - K},$$

defined on $(t_1, t_2)$, where $C = t_1 f'_+(t_1) - f(t_1)$ and $K = f'_+(t_1)$. Its derivative is given by

$$f'_{\text{left}}(x) = \frac{f''(x)(f(x) + C - Kx)}{(f'(x) - K)^2}.$$

The numerator simplifies to

$$f''(x)(f(x) + C - Kx) = \underbrace{f''(x)}_{>0} \underbrace{\left[ f(x) - (f(t_1) + f'_+(t_1)(x - t_1)) \right]}_{>0},$$

which is positive due to the convexity of $f$. This implies that shifting the right boundary $t_2$ to the left, reaching position $t_*$, also shifts the intersection point $t_{1'}$ to the left, reaching $t_{1*}$. A similar argument holds for the right boundary.

The same reasoning applies to the concave segment. $\qquad\square$

**Lemma B.9.** *Let $f_n : \mathcal{A}_f \to \mathcal{A}_g$ be continuous functions, uniformly convergent to a continuous function $f : \mathcal{A}_f \to \mathcal{A}_g$ on a compact interval $\mathcal{A}_f \subset \mathbb{R}$, and let $g_n : \mathcal{A}_g \to \mathcal{A}$ be continuous functions, uniformly convergent to a continuous function $g : \mathcal{A}_g \to \mathcal{A}$ on a compact interval $\mathcal{A}_g \subset \mathbb{R}$. Then the sequence of composition functions $g_n(f_n(x))$ converges uniformly to $g(f(x))$ on $\mathcal{A}_f$ with $n \to \infty$.*

*Proof.* An outer limit function, $g$, is uniformly continuous by the Heine–Cantor theorem (Rudin, 1976), since it is continuous and defined on the compact set $\mathcal{A}_g$. That is, for any $\epsilon_1 > 0$, there exists $\epsilon_2 > 0$ such that

$$|g(y_1) - g(y_2)| < \frac{\epsilon_1}{2}, \quad \text{whenever } y_1, y_2 \in \mathcal{A}_g \text{ and } |y_1 - y_2| < \epsilon_2.$$

Since the sequence $\{f_n(x)\}$ converges uniformly to $f(x)$ on $\mathcal{A}_f$, and $\{g_n(y)\}$ converges uniformly to $g(y)$ on $\mathcal{A}_g$, we can conclude that for any $\epsilon_1 > 0$ and $\epsilon_2 > 0$, there exists $N \in \mathbb{N}$ such that for all $n \ge N$, we simultaneously have

$$|g_n(y) - g(y)| < \frac{\epsilon_1}{2}, \qquad\qquad \text{for all } y \in \mathcal{A}_g,$$
$$|f_n(x) - f(x)| < \epsilon_2, \qquad\qquad \text{for all } x \in \mathcal{A}_f.$$

Since the range of possible values of $f_n(x)$ coincides with the range of values of $f(x)$, which equals $f_i(\mathcal{A}_f) = \mathcal{A}_g$, the domain of $g$, we obtain

$$
\begin{aligned}
|g_n(f_n(x)) - g(f(x))| &= |g_n(f_n(x)) - g(f_n(x)) + g(f_n(x)) - g(f(x))| \\
&\leq \underbrace{|g_n(f_n(x)) - g(f_n(x))|}_{\text{uniform convergence of the outer}} + \underbrace{|g(f_n(x)) - g(f(x))|}_{\text{uniform continuity of the outer}} \\
&< \frac{\epsilon_1}{2} + \frac{\epsilon_1}{2} = \epsilon_1
\end{aligned}
$$

Thus, the uniform convergence of the composition follows.

$\square$

**Lemma B.10.** *Let $f_n^i : \mathcal{A}_i \to \mathbb{R}$ be continuous functions that converge uniformly to continuous functions $f^i : \mathcal{A}_i \to \mathbb{R}$ on compacts $\mathcal{A}_i \subset \mathbb{R}$, and let $\alpha_i \in \mathbb{R}$ be constants for $i = 1, \ldots, n_l$. Then a linear combination of sequences, $f_{\boldsymbol{\alpha},n} = \sum_i^{n_l} \alpha_i f_n^i$, defined on the direct product $\mathcal{A} = \mathcal{A}_1 \times \ldots \times \mathcal{A}_{n_l} \subset \mathbb{R}^{n_l}$, converges uniformly to $f_{\boldsymbol{\alpha}} = \sum_i^{n_l} \alpha_i f^i : \mathcal{A} \to \mathbb{R}$, where $\boldsymbol{\alpha} = (\alpha_1, \ldots, \alpha_{n_l})$; that is,*

$$
\sup_{\boldsymbol{x} \in \mathcal{A}} |f_{\boldsymbol{\alpha},n}(\boldsymbol{x}) - f_{\boldsymbol{\alpha}}(\boldsymbol{x})| \xrightarrow[n \to \infty]{} 0.
$$

*The images of $f_{\boldsymbol{\alpha},n}(\mathcal{A})$ and $f_{\boldsymbol{\alpha}}(\mathcal{A})$ are compact.*

*Proof.* Since $f_{\boldsymbol{\alpha},n}(\mathcal{A})$ and $f_{\boldsymbol{\alpha}}(\mathcal{A})$ are linear combinations of continuous functions, defined on compact sets, they are continuous functions. Also, since a direct product of compact sets is compact, by the continuous mapping theorem (Rudin, 1976), the images of $f_{\boldsymbol{\alpha},n}(\mathcal{A})$ and $f_{\boldsymbol{\alpha}}(\mathcal{A})$ are also compact.

If every sequence of functions $\{f_n^i(x^i)\}$ converges uniformly to the corresponding limit function $f^i(x^i)$, then for any $\epsilon > 0$, there exists an integer $N \in \mathbb{N}$ such that for all $n \geq N$, we have

$$
\sup_{x^i \in \mathcal{A}_i} |f_n^i(x^i) - f^i(x^i)| < \frac{\epsilon}{\max_i \{\alpha_i\} n_l}
$$

for all $i = 1, \ldots, n_l$. Consequently, the supremum of the differences in the linear combination can be bounded as

$$
\begin{aligned}
\sup_{\boldsymbol{x} \in \mathcal{A}} |f_{\boldsymbol{\alpha},n}(\boldsymbol{x}) - f_{\boldsymbol{\alpha}}(\boldsymbol{x})| &= \sup_{\boldsymbol{x} \in \mathcal{A}} \left| \sum_{i=1}^{n_l} \alpha_i (f_n^i(x^i) - f^i(x^i)) \right| \\
&\leq \sum_{i=1}^{n_l} |\alpha_i| \sup_{x^i \in \mathcal{A}_i} |f_n^i(x^i) - f^i(x^i)| < \epsilon
\end{aligned}
$$

This establishes the uniform convergence of the linear combination of functions.

$\square$

**Lemma B.11.** *Let $\overline{f}^i, \underline{f}^i, f^i : \mathcal{A}_i \to \mathbb{R}$ be continuous functions on compact intervals $\mathcal{A}_i \subset \mathbb{R}$, satisfying*

$$
\underline{f}^i(x^i) \leq f^i(x^i) \leq \overline{f}^i(x^i)
$$

*for all $x^i \in \mathcal{A}_i$ and for every $i = 1, \ldots, n_l$.*

*Suppose that the index sets $I$ and $J$ are disjoint and their union forms the full sequence:*

$$
I \cup J = \{1, \ldots, n_l\}.
$$

*Then, for any coefficients $\alpha_i, \beta_j \geq 0$ for $i \in I$, $j \in J$, the following inequality holds:*

$$
\sum_{i \in I} \alpha_i \underline{f}^i(x^i) - \sum_{j \in J} \beta_j \overline{f}^j(x^j) \leq \sum_{i \in I} \alpha_i f_i(x^i) - \sum_{j \in J} \beta_j f_j(x^j) \leq \sum_{i \in I} \alpha_i \overline{f}^i(x^i) - \sum_{j \in J} \beta_j \underline{f}^j(x^j),
$$

*for all $x^i \in \mathcal{A}_i$ and $x^j \in \mathcal{A}_j$.*

*Proof.* For any non-negative coefficients $\alpha_i, \beta_j$ and given that $\underline{f}_i(x^i) \leq f_i(x^i) \leq \overline{f}_i(x^i)$, we derive the following inequalities:

$$\alpha_i \underline{f}^i(x^i) \leq \alpha_i f^i(x^i) \leq \alpha_i \overline{f}^i(x^i),$$

$$-\beta_j \overline{f}^j(x^j) \leq -\beta_j f^j(x^j) \leq -\beta_j \underline{f}^j(x^j).$$

Summing these inequalities over all indices completes the proof. □

**Lemma B.12.** *Let $\overline{f}, \underline{f}, f : \mathcal{A} \to \mathcal{B}$ be continuous functions on a compact interval $\mathcal{A} \subset \mathbb{R}$, satisfying*

$$\underline{f}(x) \leq f(x) \leq \overline{f}(x) \quad \text{for all } x \in \mathcal{A}.$$

*Furthermore, let $\overline{g}, \underline{g}, g : \mathcal{B} \to \mathbb{R}$ be continuous and monotonically increasing functions on a compact interval $\mathcal{B} \subset \mathbb{R}$, satisfying*

$$\underline{g}(y) \leq g(y) \leq \overline{g}(y) \quad \text{for all } y \in \mathcal{B}.$$

*Then, for all $x \in \mathcal{A}$, the following inequality holds:*

$$\underline{g}(\underline{f}(x)) \leq g(f(x)) \leq \overline{g}(\overline{f}(x)).$$

*Proof.* By assumption, for any $y \in \mathcal{B}$,

$$\underline{g}(y) \leq g(y) \leq \overline{g}(y).$$

Since $\underline{g}(y), g(y)$, and $\overline{g}(y)$ are monotonically increasing, it follows that for any $y_1, y_2 \in \mathcal{B}$ such that $y_1 \leq y \leq y_2$,

$$\underline{g}(y_1) \leq \underline{g}(y) \leq g(y) \leq \overline{g}(y) \leq \overline{g}(y_2).$$

Setting $y = f(x)$, $y_1 = \underline{f}(x)$, and $y_2 = \overline{f}(x)$, and using that $\underline{f}(x) \leq f(x) \leq \overline{f}(x)$ for all $x \in \mathcal{A}$, we obtain the desired result:

$$\underline{g}(\underline{f}(x)) \leq g(f(x)) \leq \overline{g}(\overline{f}(x)).$$

□

## B.2. Proof of Theorem 3.14

We establish the uniform monotonic convergence of the bounds by considering several key properties.

First, we analyze the approximation of neuron domains. Using the concept of over-approximating the input domain of each neuron (as in IBP (Gowal et al., 2019)), we define $\overline{\Omega}_i$ as a superset of the true input domain $\Omega_i$ for each neuron $i$. It is well known that if a sequence of approximations converges uniformly on $\overline{\Omega}_i$, it must also converge uniformly on any subset of $\overline{\Omega}_i$, including $\Omega_i$. Therefore, we perform all neuron-wise approximations over the supersets of their original domains.

By Lemma B.2, both the upper and lower bounds for each neuron in every layer are continuous functions over a bounded domain. Additionally, by Lemma B.5, the images of these bounds coincide with the image of the activation function. Since the activation function and its estimators are continuous, and the domain is compact, the output range remains compact throughout the approximation process. This compactness follows from the continuity of the mapping (Rudin, 1976).

Furthermore, the error between the linear interpolation (or piecewise tangent approximation) and the original activation function converges to zero as the approximation grid is refined, as established in Theorems B.6 and B.7. Consequently, the sequence of bounds on the activation function converges uniformly to the target activation function for all neurons in the network.

By Lemma B.5, the image of each activation function is preserved by both the upper and lower bounds, ensuring that the domain of uniform convergence for the bounds of the outer functions is also preserved. Moreover, by Lemma B.10, the linear combination of these estimators converges uniformly to the corresponding linear combination of the true activation functions, thereby preserving the image of the original linear combination. As a result, Lemma B.9 guarantees the uniform convergence of the composed bounds on the outer functions and the linear combinations of the inner functions.

Next, we establish monotonicity. By Lemma B.8, the sequences of upper and lower bounds on the activation functions are monotonic: the upper bounds $\{\overline{f}_n(x)\}$ are monotonically decreasing, while the lower bounds $\{\underline{f}_n(x)\}$ are monotonically

increasing. Additionally, by construction, these sequences are bounded by the target function itself, ensuring they remain within the correct range. By Lemma B.11, the linear combination of neurons' estimators forms the overall upper and lower bounds for input arguments in subsequent layers.

Moreover, the compositions of the upper and lower bounds for the inner and outer functions provide valid upper and lower bounds for the composition of the target inner and outer activation functions. Since the outer activation function (and consequently its bounds, by Lemma B.3) is non-decreasing, Lemma B.12 ensures these bounds are valid. Finally, these bounds converge monotonically by Lemmas B.8 and B.12.

Combining all the results above, we conclude that the sequences of upper and lower ReLU bounds for the network converge uniformly and monotonically to the target network. The monotonicity of the sequences, the uniform convergence of individual approximations, and the preservation of continuity and compactness through compositions collectively ensure the uniform convergence of the entire network. This completes the proof.

## C. Subnetwork equivalent for specific functions for upper/lower approximation

### C.1. Square

The quadratic function $x^2$ is not monotone on an arbitrary interval. But $x^2$ is monotonic on $\mathbb{R}_{\geq 0}$. We can modify the form of the function by representing it as a subnetwork to be a valid set of sequential monotonic operations. Since $x^2 = |x|^2$, $x \in \mathbb{R}$, and the output range of $|x|$ is exactly $\mathbb{R}_{\geq 0}$, we represent $|x|$ as a combination of monotonic ReLU functions, $|x| = \text{ReLU}(x) + \text{ReLU}(-x)$. The resulting subnetwork is drawn in Figure 3a.

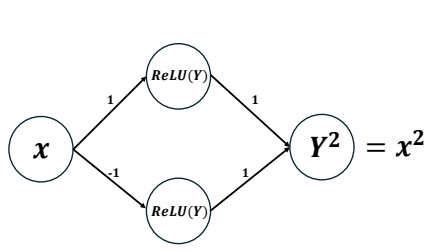

**(a)** Subnetwork equivalent to operation of taking a square.

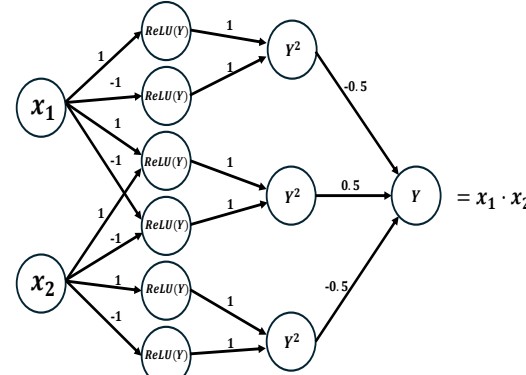

**(b)** Subnetwork equivalent to the product operation.

**Figure 3:** Subnetworks for square and multiplication operations.

### C.2. Product of two values

To find a product of two values $x_1$ and $x_2$ one can use the formula $x_1 \cdot x_2 = 0.5 \cdot ((x_1 + x_2)^2 - x_1^2 - x_2^2)$. This leads us to the feedforward network structure in Figure 3b.

### C.3. Maximum of two values

The maximum operation can be expressed via a subnetwork with ReLU activation functions only, as follows. Observing that $\max\{x_1, x_2\} = 0.5 \cdot (x_1 + x_2 + |x_1 - x_2|)$ results in the corresponding network structure in Figure 4.

### C.4. Softmax

The function $\text{softmax}$ transforms a vector of real numbers to a probability distribution. That is, if $\mathbf{x} = (x_1, \ldots, x_n) \in \mathbb{R}^n$, then there is a multivariate function $SfMax : \mathbb{R}^n \to \mathbb{R}^n$, so that

$$SfMax_i = \text{softmax}(x_i) = \frac{e^{x_i}}{\sum\limits_{j=i}^{n} e^{x_j}}$$

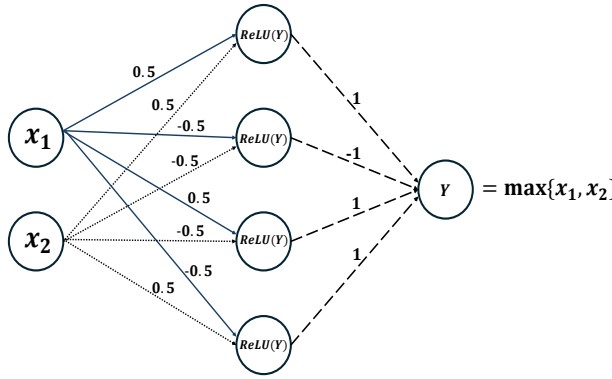

**Figure 4:** Subnetwork equivalent to a maximum of two values.

Then, $log(SfMax_i) = x_i - \log \sum_{j=i}^{n} e^{x_j}$, which is a composition of monotonic functions. This leads to the feedforward network structure in Figure 5.

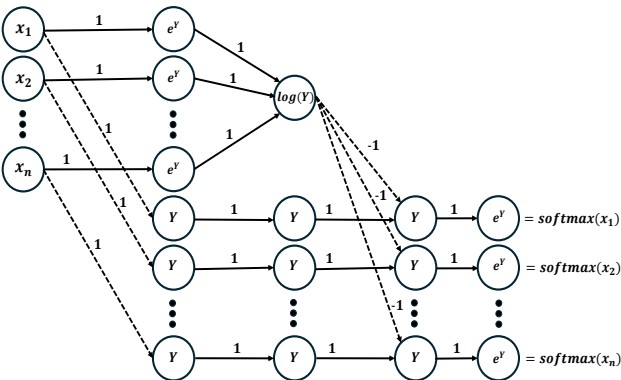

**Figure 5:** Subnetwork equivalent to one softmax node.

## D. Description of Iris Experiments

We trained a fully connected $[3 \times 12]$ ReLU NN with a final $[1 \times 3]$ linear layer, as well as a fully connected $[3 \times 12]$ $\tanh$ NN with the same final $[1 \times 3]$ linear layer, on the Iris dataset. The networks classify objects into three classes: *Setosa*, *Versicolor*, and *Virginica*, using two input features: *Sepal Length* and *Sepal Width*. The allocation of the data for these two variables in the three classes is shown in Figure 6a. The input data were rescaled to be within the interval $[0, 1]$.

**Experiment 1: ReLU-Based Network with Random Inputs.** The ReLU network was pre-trained. We next introduced randomness to the input variables by modeling them as Beta-distributed with parameters $(2, 2)$ and $(3, 2)$, respectively. The pdfs of these input distributions are shown in Figure 6b. The first is symmetric about 0.5 and the second is left-skewed.

In our first experiment, Example 3.10, we computed the exact cdf of the first output neuron (out of three) in the ReLU network before applying the softmax function. Due to the presence of a final linear layer, the output may contain negative values. To validate our computation, we compared it against a conditional *ground truth* obtained via extensive Monte Carlo simulations, where the empirical cdf was estimated using $10^5$ samples. As shown in Figure 1, both cdf plots coincide. The cdf values were computed at 100 grid points across the estimated support of the output, determined via the IBP procedure.

For further comparison, Figure 6c presents an approximation of the output pdf based on the previously computed cdf values. This is compared to a histogram constructed from Monte Carlo samples. Additionally, we include a Gaussian kernel density estimation (KDE) plot obtained from the sampled data using a smoothing parameter of $h = 0.005$. The results indicate that

our pdf approximation better represents the underlying distribution compared to KDE and tracks the histogram more closely.

**Experiment 2: Bounding a Tanh-Based Network with ReLU Approximations.**  In our second experiment, we used a pre-trained NN with a similar structure but replaced the ReLU activation functions in the first three layers with *tanh*. To approximate this network, we constructed two bounding fully connected NNs—one upper and one lower—using only ReLU activations.

We conducted computations in two regimes: one using 5 segments and another using 10 segments, into which both convex and concave regions of the *tanh* activation function at each neuron in the first three layers were divided. Over each segment, we performed piecewise linear approximations according to the procedure described in Section 3.2 and combined these approximations into three-layer ReLU networks with an additional final linear layer for both upper and lower bounds. The results of these approximations are shown in Figure 2. In the 10-segment regime, both the upper and lower approximations closely align with the original NN's output.

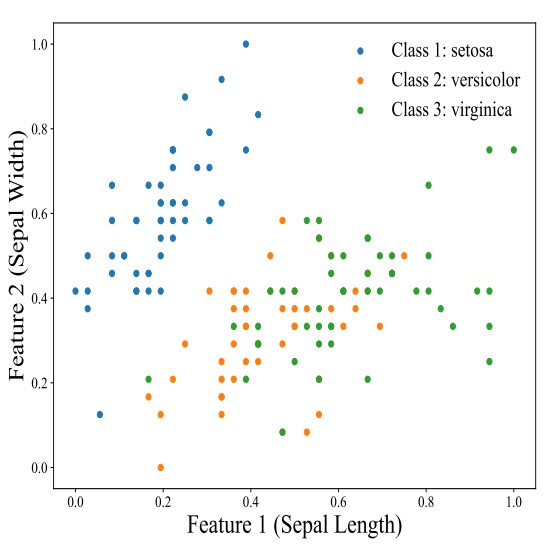

**(a)** Plot of *Sepal Width* vs *Sepal Length* with class indication.

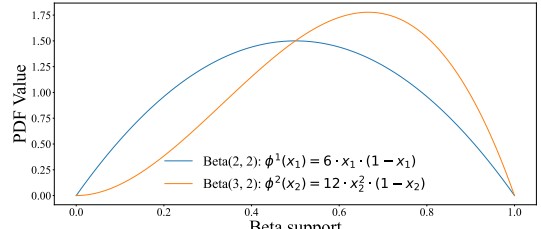

**(b)** Plots of the $Beta(2,2)$ and $Beta(3,2)$ pdfs, resp.

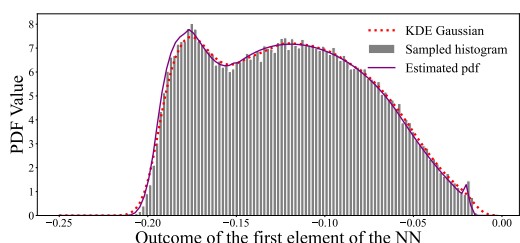

**(c)** Estimated output pdf compared with KDE with smoothing parameter $h = 0.005$ and histogram of MC simulations.

**Figure 6:** Iris Dataset

