# OpenReview forum: "Exact Upper and Lower Bounds for the Output Distribution of Neural Networks with Random Inputs"
_ICML.cc/2025/Conference — ICML 2025 poster_

### Official Review · Reviewer_WtAa · 2025-03-14

**Overall Recommendation:** 1

**Summary:**

This paper provides upper and lower bounds on the cdf of a ReLU neural network which converge to the exact cdf as the granularity increases. Other monotonic piecewise differentiable activation functions can in turn be approximated using ReLU activation functions, extending the results. Experiments validate the theoretical results.

**Claims And Evidence:**

The theoretical claims are adequately proved.

**Essential References Not Discussed:**

Even though [Krapf 24] is cited, it appears that the relationship between Theorem 2 in [Krapf 24] and Theorem 3.8 in this work is not discussed.

**Experimental Designs Or Analyses:**

See "Methods And Evaluation Criteria".

**Methods And Evaluation Criteria:**

There is a lack of detail provided about the experiments, making it difficult to evaluate the methodology. For instance, how much compute/granularity was provided for the MC, PLT, and U/L methods? Why should numerical estimation inaccuracy affect PLT but not the proposed U/L method? How many points were tested for the OOB calculations (is it the grid size)?

**Other Comments Or Suggestions:**

N/A

**Other Strengths And Weaknesses:**

This paper extends prior work on computing cdfs to providing upper and lower bounds, instead of just an approximation, which better quantifies the uncertainty of the estimate.

**Questions For Authors:**

1. Why use Theorem 3 in [Krapf 24] as a comparison for theoretical and experimental results, when Theorem 2 also applies?
2. Is there a major difference between Theorem 3 in [Krapf] and the upper/lower bounds provided? Looking at equation 6, it appears that the upper and lower bounds follow from just upper and lower bounding the integrand.

**Relation To Broader Scientific Literature:**

This paper is very much related to [Krapf 24]. One of the main theorems, Theorem 3.8, appears to be subsumed be Theorem 2 in Krapf.


Krapf, T., Hagn, M., Miethaner, P., Schiller, A., Luttner, L., & Heinrich, B. (2024). Piecewise Linear Transformation – Propagating Aleatoric Uncertainty in Neural Networks.

**Theoretical Claims:**

I did not carefully check the correctness, but the results are unsurprising in light of prior work.

---

> ### Author Rebuttal · Authors · 2025-04-01
>
> *Relation To Broader Scientific Literature and Questions For Authors:*
>
> Our approach aims to approximate as accurately as possible the **cdf** of the output in a probabilistic NN. Krapf et al (2024) estimate the **pdf** of a NN output. Our Theorem 3.8 derives the exact cdf (not pdf) of a ReLU NN when the input's pdf is a piecewise polynomial. That is, given a ReLU NN and an input with piecewise polynomial pdf, our tool (implementing Theorem 3.8) computes the cdf values **exactly** (not approximately). Theorem 2 in Krapf requires the input distribution be defined as what they call a piecewise pdf (different pdfs on the partition sets of the input support). The two theorems are *not* directly analogous.
> Furthermore, Theorem 2 in Krapf et al (2024) gives the formula of the probability density function (pdf) of the output of a feedforward NN subject to random perturbation of the inputs. The formula (Eqn (2) in Krapf et al (2024)) contains an integral that is never explained how to compute. In effect, Krapf et al (2024) do not apply Theorem 2 in their paper but rather its corollary (Theorem 3 in Krapf et al) which is a discrete approximation of the integral in Theorem 2. The latter is the reason we compare our approach to Theorem 3 in Krapf et al, which we could actually apply using the code in the Krapf et al GitHub page.
> To make things clear, in our Example 3.10, Krapf et al's method cannot compute the exact output pdf in finite time. Our approach computes the exact cdf of the output in less than a second.
>
> Regarding question 2, Theorem 3 and our approach yield similar results in the sense that the estimated cdf lies within our bounds (once the pdf is transformed to a cdf) provided the input pdf is very smooth (not wiggly).
>
> *Methods And Evaluation Criteria:*
>
> For MC, 100,000,000 sample points were used, which far exceeds the sample size in Krapf et al. The triangulation of the domain was upper bounded by 50,000 points, an ad hoc selection. The grid size we used was 20 raised to the number of classes of the output values. In the case of continuous outcome, the grid size was 20. For the Iris and Wine datasets, it was 20^3. We applied the provided PLT code from Github. For example, for the Iris dataset, we follow the settings in Krapf et al. in order to compare the performance of our method with theirs and MC. We report here computing details when we use 1, 2, 3 input variables. The emerging pattern is that our method is far more accurate and typically faster than PLT. We would also like to thank the referee for giving our method a concise name, namely U/L.
>
> *1 input variable: 25% of the data are imputed*
>
> **Computed MC in 1.873 sec**
>
> **Computed ours (U/L) in 5.700 sec**
>
> **Computed TLP in 0.348 sec**
>
> PLT out of bounds count = 2179 / 8000
>
> MC out of bounds count = 378 / 8000
>
> *2 Input variables: 25% of the data are imputed*
>
> **Computed MC in 2.114 sec**
>
> **Computed ours (U/L) in 19.663 sec**
>
> **Computed TLP in 46.043 sec**
>
> PLT out of bounds count = 0 / 8000
>
> MC out of bounds count = 53 / 8000
>
> *3 Input variables: 50% of the data are imputed*
>
> **Computed MC in 2.309 sec**
>
> **Computed ours (U/L) in 97.876 sec**
>
> **Computed TLP in 240.542 sec**
>
> PLT out of bounds count = 7 / 8000
>
> MC out of bounds count = 9 / 8000

---

### Official Review · Reviewer_xfm3 · 2025-03-23

**Overall Recommendation:** 3

**Summary:**

The paper addresses the challenge of uncertainty quantification in neural network (NN) outputs by deriving exact upper and lower bounds for the cumulative distribution function (CDF) of NN outputs under noisy (stochastic) inputs. The method is designed to apply to feedforward NNs and convolutional NNs (CNNs) using continuous monotonic piecewise differentiable activation functions, such as ReLU, tanh, and softmax. A novel aspect is the use of ReLU NNs to bound general NNs, enabling the computation of these bounds, which converge to the true CDF as resolution increases.

**Claims And Evidence:**

I find a lot of the proof details missing. For example, the proof of Theorem 3.8 is hand-wavy, and I don't see all the steps. Even more problematic is the proof of Theorem 3.11, where the authors only provide proof sketches. I don't know precisely how all these fall into place without knowing the details.

**Essential References Not Discussed:**

NA.

**Experimental Designs Or Analyses:**

No issues.

**Methods And Evaluation Criteria:**

yes

**Other Comments Or Suggestions:**

Please provide the proof details to the point where one can verify the correctness of the claims.

**Other Strengths And Weaknesses:**

See summary.

Weaknesses:

See my complain about the paper. Overall, I don't find any details to verify the proofs. Even the statements themselves look trivial to me. For example, 3.13 just seems to be a direct corollary of the classical universal theory of approximation of NN. Theorem 3.11 is perhaps the most interesting of the three theorems, but I don't find details to understand this for myself.

**Questions For Authors:**

I would want to understand each theorem-by-theorem, why the statement is non-trivial, why previous works could not solve this, and what the key difficulty addressed by this work is.

**Relation To Broader Scientific Literature:**

Not sure. See weakenesses.

**Theoretical Claims:**

yes. See my comment above about the proofs. The proofs in these papers are just not explicitly written out, and the quality of the proofs is something that I would not even tolerate in an undergraduate proof-based class, let alone a research paper.

While I understand the proofs are mostly analytical and require calculation, the authors must do this to the point where it is clear how the proofs follow. If they use any well-known theorems as black boxes, then they must explicitly cite a formal statement used, or even better include it as a preliminary subsection of the appendix, and invoke it from here.

---

> ### Author Rebuttal · Authors · 2025-04-01
>
> *Claims And Evidence and Theoretical Claims:*
>
> The proofs are not detailed as most of the steps are based on well known calculus facts and are easy to obtain. The challenge is that they are numerous and this is why we provided only the sketch of the proof. We appreciate that for the sake of clarity and completeness all the details should be made available and we will include them in the appendix if we are given the opportunity to revise. We will also place them in the arXiv version of the paper (available after the final decision).
>
> *Weaknesses:*
>
> Theorem 3.8 derives the exact cdf of a ReLU NN when the input's pdf is a piecewise polynomial. Theorem 3.11 follows from the universal approximation theorem, but the proof we provide **constructs** the sequences whereas the UAT simply shows their existence. Theorem 3.13 is our main result as it constructs bounds for the cumulative distribution function of the output of any feed forward NN whose inputs are continuous random variables with compact support. The proof of Theorem 3.13 is provided in detail. We fail to see how it is a direct corollary of the UAT and we would be interested to see the justification of this claim.

---

> > ### Comment · Reviewer_xfm3 · 2025-04-08
> >
> > Thanks for the clarifications! I missed the last line about constructiveness in Theorem 3.13.
> >
> > The presentation here should be significantly improved. The constructiveness should be an essential part of the presentation throughout. Namely,
> > 1) The real constructiveness comes from Theorem 3.11, and it is completely missing from the statement. This should be included formally.
> > 2) The constructiveness is just described in Section 3.2 as a discussion. There should be a formal algorithm box with a clear algorithmic step-by-step procedure at an appropriate level of detail, and this should be referred to when formalizing the constructiveness of Theorem 3.11.
> > 3) Similarly, in Theorem 3.13, the constructiveness should not be just written in a way that feels like a side remark. It should be emphasized and formalized by referring to some algorithmic description for this theorem as well.
> > 4) Finally, proofs should be more detailed, as I mentioned.
> >
> > I believe (1),(2),(3) above are important changes that authors must make to reflect the new ideas developed in this paper. I am raising my score to 3 and would be happy to have this paper accepted conditionally on this.

---

> > > ### Author Response · Authors · 2025-04-09
> > >
> > > We appreciate the reviewer's suggestions and the increase in the score. Should we be given the opportunity to revise, we can easily address all points raised.

---

### Official Review · Reviewer_3TKr · 2025-03-24

**Overall Recommendation:** 4

**Summary:**

This paper proposes a novel method to compute exact upper and lower bounds for the cdf of a neural network’s output, assuming stochastic inputs. Key contributions include:
- A method to compute the exact cdf of the output of ReLU networks under inputs with piecewise polynomial pdf over compact hyperrectangle.
- A method to compute tight upper and lower bounds of the cdf of the output of neural networks with piecewise differentiable activations under inputs with a pdf with compact support.

**Claims And Evidence:**

Correct me if I misunderstood but it seems that Table 1 merely shows that MC and PLT disagree with your bounds, but it does not provide evidence to support the correctness of your bounds?

In Figure 1, it says that you are assuming beta-distributed inputs, but how can you assume that if the input is the Iris dataset? Are you talking about random perturbations to the Iris dataset?

**Essential References Not Discussed:**

Not that I know of

**Experimental Designs Or Analyses:**

see claims and evidence

**Methods And Evaluation Criteria:**

see claims and evidence

**Other Comments Or Suggestions:**

see other comments

**Other Strengths And Weaknesses:**

Strengths: strong theoretical contribution
Weaknesses: see other comments

**Questions For Authors:**

if you could address my concerns about the experiments, and answer my comments on the theoretical contributions, I would be willing to increase my score.

**Relation To Broader Scientific Literature:**

This paper contributes to an important area of the literature on estimating uncertainty in neural networks.

**Theoretical Claims:**

I did not check the correctness of the proofs, only looked at the statements.

Theorem 3.13 seems to be a strong result but it is not well presented. For any F (y), of course there always exists upper and lower bounds F_n such that they converge to F (y): the interesting part of the result is that the authors are talking about particular F_n that can be constructed, so the last sentence "The sequence {F n} can be constructed by bounding the distributions of
sequences of ReLU NNs." should be the core of the theorem. The subsequent two paragrpahs (l.289-304 right block) where the authors describe how to construct F_n should be part of the statement, and not using words but mathematical notations... Also, from the text in the second paragraph (l.295-304 right block), it seems that polynomial pdfs are never used, so it would be perhaps be less confusing if Theorem 3.8 was stated for piecewise constant pdfs.

I'm wondering if Theorem 3.11 does not follow directly from universal approximation theorems?

---

> ### Author Rebuttal · Authors · 2025-04-01
>
> *Theoretical Claims:*
>
> You are right that Theorem 3.11 follows from the universal approximation theorem. But it also **constructs** the sequences whereas the UAT simply shows their existence.
> Indeed the significance of Theorem 3.13 is that the bounds are built using ReLU functions, which is our main innovation as it also renders our approach applicable to general feedforward NNs. If we have the opportunity to revise, we will restate the theorem as you suggest.
> Theorem 3.8 obtains a more general result even though we apply it in the proof of Theorem 3.13 only with piecewise constant pdfs.
>
> *Claims And Evidence:*
>
> In Table 1 we see that our bounds differ from the competing PLT and MC methods. You are right that it provides no evidence for the correctness of ours. We show formally that our bounds are correct. Table 1 serves to illustrate that the other two methods, PLT and MC, are less accurate. In effect, if we *significantly reduce* the accuracy of our bounds then they include the PLT and MC approximations. In other words, we can recover the PLT and MC estimates by making our method less precise.
> In the Iris data, the inputs are perturbed by Beta noise since in this case (the Beta pdf is a polynomial) we can compute exact bounds. Of course, we could have had used other random perturbations as well.

---

> > ### Comment · Reviewer_3TKr · 2025-04-03
> >
> > I am inclined to accept this paper as the authors have promised to address my concerns in their revision. The paper provides a strong theoretical contribution that results an efficient computational method to compute bounds of the cdf of the output of neural networks that are tighter than existing methods. I have raised my score.

---

> > > ### Author Response · Authors · 2025-04-09
> > >
> > > We would like to thank the reviewer for his comments on our work and for increasing his score.

---

### Decision · Program_Chairs · 2025-05-01

**Decision:**

Accept (poster)

**Comment:**

The paper looks at estimation of CDF for neural networks with
stochastic inputs. This is applied to ReLU and some other
activations. The topic is of interest in uncertainty quantification.
Most of reviewers agree that the contribution is novel and has
theoretical potential.  At the same time, some reviewers are concerned
with lack of experimental detail and insufficient discussion regarding
relation to [Krapf 24].  Some reviewers also point out lack of detail,
rigor, and clarity in the proofs.  Authors are strongly encouraged to
address these limitations.